# RAS–p110α signalling in macrophages is required for effective inflammatory response and resolution of inflammation

**Alejandro Rosell[1†], Agata Adelajda Krygowska[2†], Marta Alcón Pérez[1], Cristina Cuesta[1], Mathieu-Benoit Voisin[3], Juan de Paz[1], Héctor Sanz-Fraile[4], Vinothini Rajeeve[5], Ana Carreras-González[6], Alberto Berral-González[7], Ottilie Swinyard[2], Enrique Gabandé-Rodríguez[2], Julian Downward[8], Jordi Alcaraz[4,9], Juan Anguita[7,10,11], Carmen García-Macías[11], Javier De Las Rivas[6], Pedro R Cutillas[5], Esther Castellano Sanchez[1,2]***

[1]Tumour-Stroma Signalling Lab., Centro de Investigación del Cáncer, Instituto de Biología Molecular y Celular del Cáncer, Consejo Superior de Investigaciones Científicas (CSIC)-Universidad de Salamanca, Campus Miguel de Unamuno, Salamanca, Spain; [2]Centre for Cancer and Inflammation, Barts Cancer Institute, Queen Mary University of London, London, United Kingdom; [3]Centre for Microvascular Research, William Harvey Research Institute, Queen Mary University of London, London, United Kingdom; [4]Unit of Biophysics and Bioengineering, Department of Biomedicine, School of Medicine and Health Sciences, Universitat de Barcelona, Barcelona, Spain; [5]Centre for Cancer Genomics and Computational Biology, Cell Signalling and Proteomics Laboratory, Barts Cancer Institute, Queen Mary University of London, London, United Kingdom; [6]Bioinformatics and Functional Genomics, Centro de Investigación del Cáncer, Instituto de Biología Molecular y Celular del Cáncer, Consejo Superior de Investigaciones Científicas (CSIC)-Universidad de Salamanca, Salamanca, Spain; [7]Inflammation and Macrophage Plasticity Lab, CIC bioGUNE, Derio, Spain; [8]Oncogene Biology Laboratory, Francis Crick Institute, London, United Kingdom; [9]Institute for Bioengineering of Catalonia (IBEC), The Barcelona Institute for Science and Technology (BIST), Barcelona, Spain; [10]Ikerbasque, Basque Foundation for Science, Bilbao, Spain; [11]Pathology Unit, Centro de Investigación del Cáncer, Instituto de Biología Molecular y Celular del Cáncer, Universidad de Salamanca, Salamanca, Spain

***For correspondence:**
ecastellano@usal.es

[†]These authors contributed equally to this work

**Competing interest:** The authors declare that no competing interests exist.

## eLife Assessment

This **useful** study investigates the impact of disrupting the interaction of RAS with the PI3K subunit p110α in macrophage function in vitro and inflammatory responses in vivo. **Solid** data overall supports a role for RAS-p110α signalling in regulating macrophage activity and so inflammation, however for many of the readouts presented the magnitude of the phenotype is not particularly pronounced. Further analysis would be required to substantiate the claims that RAS-p110α signalling plays a key role in macrophage function. Of note, the molecular mechanisms of how exactly p110α regulates the functions in macrophages have not yet been established.

**Abstract** Macrophages are crucial in the body's inflammatory response, with tightly regulated functions for optimal immune system performance. Our study reveals that the RAS–p110α signalling

pathway, known for its involvement in various biological processes and tumourigenesis, regulates two vital aspects of the inflammatory response in macrophages: the initial monocyte movement and later-stage lysosomal function. Disrupting this pathway, either in a mouse model or through drug intervention, hampers the inflammatory response, leading to delayed resolution and the development of more severe acute inflammatory reactions in live models. This discovery uncovers a previously unknown role of the p110α isoform in immune regulation within macrophages, offering insight into the complex mechanisms governing their function during inflammation and opening new avenues for modulating inflammatory responses.

## Introduction

Phosphatidylinositol 3-kinases (PI3K) are a family of lipid kinases that phosphorylate phosphatidylinositides (PtdIns) at the 3′-hydroxyl group (*Cuesta et al., 2021*). Upon activation, PI3K phosphorylates phosphatidylinositol 4,5-bisphosphate (PIP$_2$) to generate phosphatidylinositol 3,4,5-trisphosphate (PIP$_3$). PIP$_3$ serves as a second messenger that recruits proteins containing pleckstrin homology (PH) domains, such as Akt (also known as protein kinase B) (*Cantley, 2002*; *Castellano and Downward, 2011*). This activation of PI3K regulate various cellular functions, including cell proliferation, growth, survival, motility, inflammation, and metabolism, among others (*Cuesta et al., 2021*; *Madsen and Vanhaesebroeck, 2020*). In macrophages, the activation of PI3K–Akt signalling is crucial to restrict inflammation and to promote anti-inflammatory responses in Toll-like receptor-induced macrophages, contributing to macrophage polarization (*Troutman et al., 2012*; *Vergadi et al., 2017*).

PI3Ks are heterodimeric lipid kinases composed of catalytic and adaptor/regulatory subunits that can be categorized into three classes based on their structures and substrate specificities (*Castellano and Downward, 2011*; *Kok et al., 2009*). Class I catalytic isoforms, including p110α, p110β, p110γ, and p110δ, play essential roles in integrating signals from growth factors, cytokines, and other environmental cues. While p110α and p110β are ubiquitously expressed, p110δ and p110γ are largely restricted to the myeloid and lymphoid lineages (*Kok et al., 2009*; *Hawkins and Stephens, 2015*; *Okkenhaug, 2013*). While p110α is primarily associated with cell growth regulation and survival in epithelial cells, it must be considered that this isoform is also expressed in immune cells, including macrophages. The role played by p110α in macrophages is not well understood, although some studies have suggested that it might regulate the survival and the regulation of the phagocytic activity of macrophages (*Tamura et al., 2009*). In the context of cancer, impairment of RAS binding to p110α somatically results in reduced recruitment of macrophages to the tumour site (*Castellano et al., 2013*; *Murillo et al., 2014*). Additionally, this disruption leads to a change in macrophage polarization, favouring a more pro-inflammatory M1 state (*Murillo et al., 2014*). These findings suggest that p110α plays a crucial role in regulating macrophage-dependent functions. However, despite these insights, the precise impact of p110α on macrophage function and the underlying molecular mechanisms influencing the inflammatory response are not yet fully understood.

Inflammation is a complex and tightly regulated series of events triggered by various stimuli such as pathogens, harmful mechanical and chemical agents, and autoimmune reactions. The inflammatory response primarily occurs in vascularized connective tissues, involving a dynamic interplay of plasma components, circulating cells, blood vessels, and cellular and extracellular factors (*Bennett et al., 2018*; *Chaplin, 2010*; *Chen et al., 2018*; *Medzhitov, 2008*; *Medzhitov, 2021*). During inflammation, mediators released by recruited leukocytes orchestrate a response that aims to facilitate tissue repair and protect the body against harmful stimuli (*Medzhitov, 2008*; *Medzhitov, 2021*; *Gerhardt and Ley, 2015*).

Macrophages play a vital role in the inflammatory response by performing functions such as antigen presentation, phagocytosis, and immunomodulation (*Medzhitov, 2008*; *Medzhitov, 2021*). Their role begin with the active recruitment of monocytes from the bloodstream to the site of infections (*Shi and Pamer, 2011*) where they differentiate into macrophages and recognize microbes and cellular debris through specific mechanisms. Macrophages subsequently actively participate in phagocytosis, a vital process involving the internalization and elimination of pathogens. Microbial destruction predominantly takes place within lysosomes and phagolysosomes (*Ballabio and Bonifacino, 2020*; *Saftig and Puertollano, 2021*; *Gordon, 2016*; *Kourtzelis et al., 2020*). At later stages of inflammation,

macrophages contribute to the resolution of inflammation, thus preventing progression from acute to persistent inflammation that would cause additional tissue damage.

In this study, we have used a combination of cell biology techniques and animal models to better understand the role of RAS-dependent activation of p110α at the different stages of the inflammatory response. Our findings show that RAS–p110α signalling plays a key role in the initial stages of inflammation, facilitating the extravasation of monocytes from the bloodstream by promoting the necessary cytoskeletal changes. Subsequently, RAS–p110α has a crucial role in lysosomal acidification and activation of cathepsins, which are indispensable for efficient degradation of lysosomal cargo; when these functions are impaired, prolonged acute inflammatory responses and delayed resolution steps are observed. These results significantly enhance our understanding of the complex mechanisms governing the immune response to inflammation, emphasizing the pivotal role played by RAS–p110α signalling in orchestrating proper monocyte extravasation and maintaining optimal lysosomal function.

## Results

### Disruption of RAS–p110α causes prolonged and more acute responses to inflammatory stress in vivo

Previous data suggested that, in a tumoural setting, somatic disruption of RAS–p110α prevents macrophage recruitment to tumours (*Castellano et al., 2013*; *Murillo et al., 2014*) and favours polarization of macrophages to a pro-inflammatory phenotype (*Murillo et al., 2014*), suggesting a possible role for p110α in macrophage function. To determine whether RAS-dependent activation of p110α participates in innate or adaptive immune responses to inflammation in macrophages, we used an established mouse model designed for the tamoxifen-inducible disruption of RAS binding to p110α (*Castellano et al., 2013*; *Gupta et al., 2007*; *Figure 1—figure supplement 1A*). This mouse model introduces two-point mutations, T208D and K227A, in the RAS-binding domain (RBD) of the endogenous *Pik3ca* allele (*Pik3ca^RBD*), enabling the selective disruption of the RAS–p110α interaction (*Gupta et al., 2007*). Wild-type (*Pik3ca^WT*) and *Pik3ca^RBD* mice were bred with mice containing a floxed *Pik3ca* allele (*Zhao et al., 2006*), along with a strain containing a conditional Cre recombinase (Cre-ERT2) allele targeted to the ubiquitously expressed *Rosa26* locus. The resulting *Pik3ca*^WT/Flox^ and *Pik3ca*^RBD/Flox^ mice displayed no discernible phenotype and exhibited behaviour consistent with Pik3ca^WT^ mice (*Castellano et al., 2013*; *Gupta et al., 2007*). Activation of Cre-recombinase by tamoxifen led to the excision of the floxed *Pik3ca* allele, resulting in mice expressing either one *Pik3ca^WT* allele (*Pik3ca*^WT/−^) or one *Pik3ca*^RBD^ allele (*Pik3ca*^RBD/−^) (*Castellano et al., 2013*). This inducible genetic manipulation strategy allows us to selectively disrupt the RAS–p110α interaction in a controlled and temporally regulated manner, providing a valuable tool for dissecting the contributions of this pathway to immune responses in the context of inflammation.

Bone marrow-derived macrophages (BMDMs) were generated from tibias and femurs of *Pik3ca*^WT/Flox^ and *Pik3ca*^RBD/Flox^ mice, ensuring efficient removal of the floxed allele (*Figure 1—figure supplement 1B*). Subsequently, these BMDMs were induced towards a pro-inflammatory state through stimulation with lipopolysaccharide (LPS) and interferon gamma (IFN-γ). To assess the impact of RAS–p110α-binding deficiency on inflammatory intracellular signalling, we first examined Akt activation. The results revealed a decrease in Akt activation levels under inflammatory conditions in BMDMs lacking RAS–p110α binding, while no discernible change was observed in ERK activation, another well-known RAS effector (*Figure 1A*, *Figure 1—figure supplement 1C, D*). Interestingly, the decrease in Akt activation was accompanied by a decrease in the activation of NF-κB (*Figure 1B*).

Given the role of p65 in the expression of pro-inflammatory cytokines and chemokines (*Zhao et al., 2021*) we next analysed the expression of various inflammation mediators using a cytokine array in unstimulated and LPS- and IFN-γ-stimulated macrophages. Results showed that, under unstimulated conditions, RAS–PI3K disruption decreases the expression of IP-10, MIP-1α (Ccl3), JE (Ccl2/MCP1), IL-16, and IL-12p70, with no upregulated cytokines observed (*Figure 1—figure supplement 1E*). The downregulation of IP-10, MIP-1α, and JE indicates an impaired ability to recruit monocytes, macrophages, and T cells to sites of inflammation. This suggests that macrophages lacking RAS–PI3K interaction may have a reduced capacity to mount a robust immune response, particularly in recruiting and activating essential immune cells needed to combat pathogens or initiate inflammation.

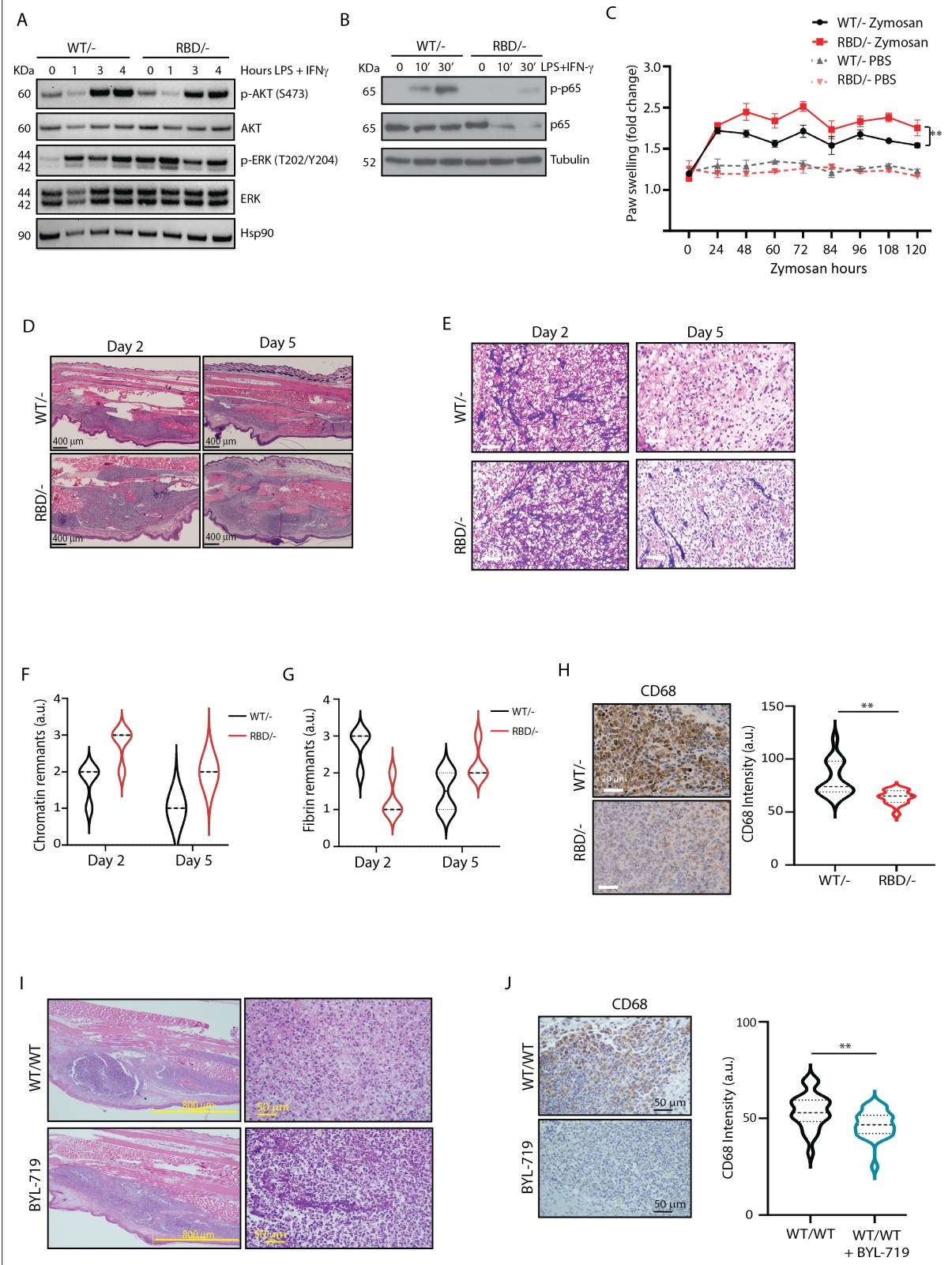

**Figure 1.** *Pik3ca^RBD/−* mice have impaired responses to inflammatory insults. (**A**) Western blotting showing activation of Akt and ERK in *Pik3ca^WT/−* and *Pik3ca^RBD/−* bone marrow-derived macrophages (BMDMs) activated towards a pro-inflammatory state with lipopolysaccharide (LPS) and interferon gamma (IFN-γ) at the indicated time points. (**B**) Western blotting showing activation of NF-κB (p65) in *Pik3ca^WT/−* and *Pik3ca^RBD/−* BMDMs activated towards a pro-inflammatory state with LPS and IFN-γ at the indicated time points. (**C**) *Pik3ca^WT/−* and *Pik3ca^RBD/−* mice were injected with zymosan or

*Figure 1 continued on next page*

*Figure 1 continued*

PBS in the back-hind paws and inflammation was measured and plotted over time, *Pik3ca^{WT/−}* PBS *n* = 6; *Pik3ca^{RBD/−}* PBS *n* = 5; *Pik3ca^{WT/−}* Zymosan *n* = 4; *Pik3ca^{RBD/−}* Zymosan *n* = 4. Error bars indicate mean ± SEM. Significance using two-way ANOVA test: **p < 0.01. (**D**) Representative haematoxylin and eosin (H&E) images of the inflamed area of *Pik3ca^{WT/−}* and *Pik3ca^{RBD/−}* paws injected with zymosan at the indicated times. (**E**) Representative images of cellularity present in the inflamed abscess of *Pik3ca^{WT/−}* and *Pik3ca^{RBD/−}* paws injected with zymosan at the indicated times. (**F**) Graph showing quantification of loose chromatin present in the inflamed abscess. (**G**) Graph showing quantification of fibrin present in the inflamed abscess. (**H**) Representative images of macrophages (CD68-positive cells) present in the inflamed abscess of *Pik3ca^{WT/−}* and *Pik3ca^{RBD/−}* paws injected with zymosan and quantification of macrophages (CD68-positive cells) present. (**I**) Representative H&E images of the inflamed area and cellularity of *Pik3ca^{WT/WT}* paws and *Pik3ca^{WT/WT}* paws treated with BYL-719 injected with zymosan. (**J**) Representative images and quantification of macrophages (CD68-positive cells) present in the inflamed abscess of *Pik3ca^{WT/WT}* paws and *Pik3ca^{WT/WT}* paws treated with BYL-719 injected with zymosan and quantification of macrophages (CD68-positive cells). Error bars indicate mean ± SEM. Significance using Student's *t* test: **p < 0.01.

The online version of this article includes the following source data and figure supplement(s) for figure 1:

**Source data 1.** Original membranes corresponding to *Figure 1A*, labelled.

**Source data 2.** Original membranes corresponding to *Figure 1A*.

**Source data 3.** Original membranes corresponding to *Figure 1A*, labelled.

**Source data 4.** Original membranes corresponding to *Figure 1B*.

**Figure supplement 1.** *Pik3ca^{RBD/−}* mice have more extended inflamed tissue after zymosan injection.

**Figure supplement 1—source data 1.** DNA gel showing a representative gel of sample genotyping.

**Figure supplement 1—source data 2.** Original gel image with no edits.

Regarding LP + IFN-γ-stimulated macrophages, results showed a decrease in the expression of IL-1β and IL-17, key drivers of pro-inflammatory responses, and an upregulation of IL-7, Ilra, JE, BLC, I-309, Eotaxin, and G-CSF (*Figure 1—figure supplement 1F*). Elevated levels of these factors are associated with enhanced chemotactic signals and regulatory functions. Thus, the cytokine and chemokine expression profile observed in RAS–PI3K-deficient macrophages suggests impaired pro-inflammatory responses and altered immune cell recruitment patterns, potentially influencing the resolution of inflammation and tissue repair processes.

Next, we assessed whether the absence of RAS binding to p110α affects the ability of BMDMs to acquire a pro-inflammatory state characterized by increased expression of markers such as CD80, CD86, and MHCII (*Biswas and Mantovani, 2010*; *Mercalli et al., 2013*). To assess this, *Pik3ca^{RBD/−}* and *Pik3ca^{WT/−}* BMDMs were stimulated with LPS + IFN-γ, followed by flow cytometry analysis. Expression levels of CD80 (*Figure 1—figure supplement 1G*), CD86 (*Figure 1—figure supplement 1H*), and MCHII (*Figure 1—figure supplement 1I*) were examined. No significant differences were observed in the expression levels of any of these markers between the two genotypes under study.

Consequently, our next objective was to investigate whether in vivo disruption of RAS–p110α might lead to altered inflammatory responses. To address this, we administered tamoxifen to 10- to 12-week-old *Pik3ca^{WT/flox}* and *Pik3ca^{RBD/flox}* mice with tamoxifen and, after a 2-week interval, conducted a paw swelling assay. In this assay, zymosan (10 μg/μl) or PBS was injected into the hind paws of *Pik3ca^{RBD/−}* and *Pik3ca^{WT/−}* mice with paw thickness measured at regular intervals over a 5-day period. The results revealed a significant increase in paw inflammation in *Pik3ca^{RBD/−}* mice compared to *Pik3ca^{WT/−}* mice, evident from the earlier time points measured and persisting throughout the experiment (*Figure 1D and E*). Paws from the *Pik3ca^{WT/flox}* and *Pik3ca^{RBD/flox}* mice, which had not received tamoxifen, exhibited comparable levels of inflammation (*Figure 1—figure supplement 1J*). This reaffirms that the absence of p110α triggers a significant alteration in the inflammatory response and confirms that *Pik3ca^{WT/flox}* and *Pik3ca^{RBD/flox}* mice do not show any phenotype, as previously described (*Castellano et al., 2013*; *Gupta et al., 2007*). Additionally, analysis of the blood sedimentation rate, a reliable indicator of systemic inflammation levels (*Paulsen et al., 2017*), showed a lower basal sedimentation ratio in *Pik3ca^{RBD/−}* mice under PBS-treated conditions (*Figure 1—figure supplement 1K*). Following zymosan injection, both genotypes exhibited an increase in sedimentation rate, but the rise was more pronounced in *Pik3ca^{RBD/−}* mice, indicating a heightened systemic inflammatory response. These findings underscore that disruption of RAS–p110α interaction results in an exacerbated inflammatory state, reflected in both localized paw inflammation and systemic inflammatory mediator levels.

To delve deeper into the inflammatory response induced by zymosan, we collected paw samples at 2- and 5-day post-injection and conducted haematoxylin and eosin studies. This approach enabled

a comprehensive analysis, allowing us not only to scrutinize the early stages of inflammation but also to monitor the subsequent resolution phase of the inflammatory process. Paws obtained from mice injected with zymosan for 2 days displayed extensive areas of damaged connective tissue in both *Pik3ca$^{RBD/-}$* and *Pik3ca$^{WT/-}$* paws (*Figure 1D*). Notably, the inflamed region in *Pik3ca$^{RBD/-}$* mice was larger compared to control samples. After 5 days of zymosan injection, there was a significant reduction in the inflamed area, although it remained comparatively larger in the paws from *Pik3ca$^{RBD/-}$* mice (*Figure 1D*). This observation suggests a prolonged and heightened inflammatory response in mice lacking RAS–p110α interaction, emphasizing the role of this interaction in the regulation of inflammatory processes.

The cellular composition within the inflammatory abscess offers crucial insights into the severity and progression of the disease. Close examination with the pathologist revealed features indicative of an acute inflammatory response, including an inflammatory abscess with elevated numbers of polymorphonuclear cells, primarily neutrophils, and macrophages displaying altered cell shape and increased cell death, accompanied by fibrin and chromatin deposition (*Figure 1E, F*). Notably, *Pik3ca$^{RBD/-}$* mice exhibited lower numbers of macrophages, a larger necrotic area with increased chromatin remnants, and reduced fibrin content compared to the paws from *Pik3ca$^{WT/-}$* mice (*Figure 1E-G*). By day 5, paws from *Pik3ca$^{WT/-}$* mice showed a significant increase in the number of macrophages, a decrease in polymorphonuclear cells, and a nearly complete resolution of the necrotic area, indicating the initiation of the resolution phase. Moreover, there were abundant activated fibroblasts, suggesting active production of new connective tissue. In contrast, paws from *Pik3ca$^{RBD/-}$* mice exhibited a delayed healing process characterized by a larger central area of polymorphonuclear cells, abundant fibrin deposits, reduced numbers of infiltrating macrophages, and limited fibroblast activity (*Figure 1E-H*). These findings collectively indicate an imbalance in the inflammatory response and a slower progression towards resolution in the absence of RAS–p110α interaction, emphasizing the pivotal role of this interaction in orchestrating an effective and timely resolution of inflammation.

To evaluate the presence of macrophages within the inflammatory lesion, we performed specific immunohistochemical analysis using the macrophage-specific marker CD68 in paws from day 5, where more cellular preservation and initiation of the healing process had been observed. A notable decrease in the number of CD68-positive cells was observed in the inflamed abscess region of *Pik3ca$^{RBD/-}$* mice (*Figure 1G*).

To further confirm the involvement of p110α signalling in the acute inflammatory response, *Pik3ca$^{WT/WT}$* mice were subjected to daily treatment with BYL719 (Alpelisib), a specific inhibitor of p110α isoform. After an initial 48-hr treatment period, the mice received injections of zymosan or PBS into their back-hind paws and were sacrificed 2 days later for analysis. Similar to what we had observed in *Pik3ca$^{RBD/-}$* mice, the inflamed area in BYL719-treated mice exhibited a larger extension than the inflamed are from non-treated mice (*Figure 1H*, *Figure 1—figure supplement 1L*). The central necrotic region was significantly larger in the BYL719-treated mice, and it contained large amount of apoptotic polymorphonuclear cells, lower number of macrophages, and increased deposits of chromatin (*Figure 1H*) resembling the observations from *Pik3ca$^{RBD/-}$* mice. Immunostaining for CD68 revealed a reduction in the number of macrophages present in the inflammatory abscess upon inhibition of p110α signalling with BYL719 treatment (*Figure 1I*).

In summary, our findings highlight that both genetic disruption of RAS–p110α interaction and pharmacological inhibition of p110α contribute to an expanded inflamed area and central necrotic region, concurrently reducing macrophage infiltration in a zymosan-induced inflammation model. These results collectively underscore the pivotal role of the p110α isoform of PI3K in orchestrating the resolution of inflammatory responses, emphasizing its significance in modulating the dynamics and outcomes of inflammatory processes.

## Disruption of RAS binding to p110α impairs the number of inflammatory monocytes in blood and spleen

Given the decrease in macrophages observed in the inflammatory abscess, we next set out to determine whether disruption of RAS binding to p110α had an effect on the number of monocytes circulating in the blood of adult mice. *Pik3ca$^{WT/flox}$* and *Pik3ca$^{RBD/flox}$* mice were treated with tamoxifen at 12–14 weeks of age and 4 weeks later, blood was collected by cardiac puncture and immune populations were analysed by flow cytometry. To assess the effects of RAS–p110α disruption on immune

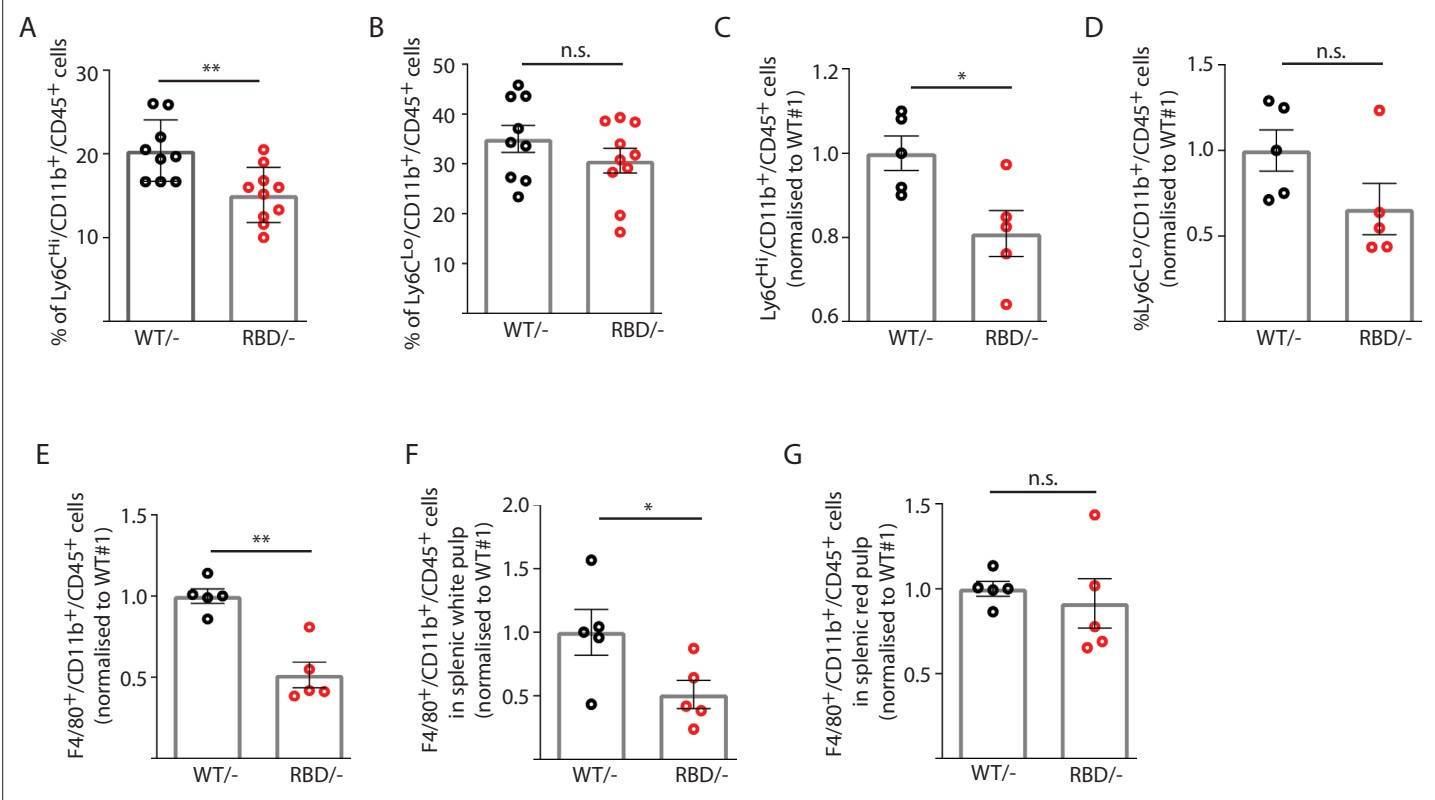

**Figure 2.** Disruption of RAS–p110α interaction decreases the number of inflammatory monocytes in blood and spleen. Twelve-week-old mice were treated with tamoxifen and after 4 weeks, flow cytometry analysis was performed to determine (**A**) inflammatory monocytes in circulating blood; (**B**) alternatively activated monocytes in circulating blood; (**C**) inflammatory monocytes in spleen; (**D**) classically activated monocytes in spleen; (**E**) macrophages in spleen; (**F**) macrophages in spleen's white pulp; (**G**) macrophages in spleen's red pulp. Data are presented as percentage of positive cells for the indicated markers. Black dots represent data from *Pik3ca^{WT/–}* mice and red dots represent data from *Pik3ca^{RBD/–}* mice. Each dot represents an individual mouse. Error bars indicate mean ± SEM. Significance using Student's *t* test: *p < 0.05; **p < 0.01.

The online version of this article includes the following figure supplement(s) for figure 2:

**Figure supplement 1.** Gating strategy for immune population analysis in blood and spleen.

**Figure supplement 2.** Disruption of RAS–p110α interaction does not affect the number of B or T cells in blood or spleen.

populations, we employed a flow cytometry gating strategy as described in the Methods section. Briefly, CD45+ cells were initially gated to define the overall immune cell population. Within this population, B cells were identified by CD19+ CD11b– staining, and T cells were defined by CD3+ expression. Further differentiation of T cell subtypes (CD4+, CD8+, and double-negative T cells) was conducted based on CD4 and CD8 markers within the CD3+ cell population. Additionally, myeloid cells were gated as CD11b+ cells, with further identification of granulocytes and monocytes based on Ly6G and Ly6C expression. Specifically, granulocytes/neutrophils were characterized as Ly6G+ Ly6C+, classical monocytes as Ly6G– Ly6C+, and alternative monocytes as Ly6G– Ly6C– (*Figure 2— figure supplement 1A*). We found a decrease in the number of circulating classical (or inflammatory) monocytes (Ly6C^{Hi}/Ly6G^{-}/CD11b^{+} cells) in *Pik3ca^{RBD/–}* mice (*Figure 2A*) and no changes were observed in non-classical (or non-inflammatory) monocytes (Ly6C^{Lo}/Ly6G^{-}/CD11b^{+} cells) (*Figure 2B*). Together with the decrease in inflammatory monocytes, we observed an increase in the number of neutrophils (Ly6C^{-}/Ly6G^{+}/CD11b^{+} cells) (*Figure 2—figure supplement 2A*). We did not detect differences in the numbers of T cells (CD3^{+}, CD8^{+}, or CD4^{+}) (*Figure 2—figure supplement 2B*) or B cells (CD19^{+}) (*Figure 2—figure supplement 2C*) in the blood after the disruption of RAS binding to p110α. We also analysed these same cell populations in *Pik3ca^{WT/flox}* and *Pik3ca^{RBD/flox}* mice and no differences were found (data not shown).

Since splenic monocytes resemble their blood counterparts (*Swirski et al., 2009*) we next aimed at determining whether splenic monocyte population would also be altered after disruption of

RAS–p110α interaction. Flow cytometry from $Pik3ca^{WT/-}$ and $Pik3ca^{RBD/-}$ mice was performed following the same gating strategy as described for blood samples (*Figure 2—figure supplement 1B*). As observed in circulating blood, the number of classical monocytes (Ly6C$^{Hi}$/Ly6G$^-$/CD11b$^+$) in the spleen decreased after disruption of RAS–p110α interaction (*Figure 2C*) and no differences were found in non-classical activated monocytes (Ly6C$^{Lo}$/ Ly6G$^-$/CD11b$^+$) (*Figure 2D*). We also checked the levels of resident macrophages present in the spleen and results showed that spleens from the $Pik3ca^{RBD/-}$ mice had a decrease in the number of differentiated macrophages (F4/80$^+$/CD11b$^+$/CD45$^+$) (*Figure 2E*). Analysis of macrophages in the white and red pulp of the spleen indicated that $Pik3ca^{RBD/-}$ mice had a significant decrease in the macrophage population in the former region (*Figure 2F, G*). No differences were found in the number of granulocytes (*Figure 2—figure supplement 2D*), B cells (*Figure 2—figure supplement 2E*), or T cells (*Figure 2—figure supplement 2F*) present in the spleen between $Pik3ca^{RBD/-}$ and $Pik3ca^{WT/-}$ mice. There were no differences in spleen size from $Pik3ca^{RBD/-}$ and $Pik3ca^{WT/-}$ mice either (*Figure 2—figure supplement 2G*).

Given the decrease in the number of inflammatory monocytes and macrophages observed in blood and spleens of $Pik3ca^{RBD/-}$ mice, we wondered if disruption of RAS activation of p110α could lead to a decrease in myeloid bone marrow precursors. Myeloid lineage descends from a common myeloid progenitor (CMP) in bone marrow and traverse into blood as mature cells (*Akashi et al., 2000*). CMP differentiates into the granulocyte–macrophage progenitor (GMP) and the megakaryocyte–erythroid progenitor (MEP). The CMP can generate all types of myeloid colonies, whereas the GMP or the MEP produces only granulocyte macrophage or megakaryocyte erythrocyte (MegE) lineage cells, respectively (*Figure 2—figure supplement 2H*). No significant differences were noted in the quantity of CMP progenitors within the bone marrow $Pik3ca^{RBD/-}$ and $Pik3ca^{WT/-}$ mice (*Figure 2—figure supplement 2I–L*). This observation suggests that the variations observed in blood and spleen parameters are not attributable to impairments in progenitor fate.

Finally, bone marrow precursors from $Pik3ca^{RBD/flox}$ and $Pik3ca^{WT/flox}$ mice were differentiated into macrophages in vitro, and the number of BMDMs was determined by flow cytometry analysis. No differences were found in the number of BMDMs obtained from $Pik3ca^{RBD/-}$ and $Pik3ca^{WT/-}$ mice (*Figure 2—figure supplement 2M*), suggesting that the disruption of RAS–p110a signalling do not interfere with the ability of bone marrow precursors to differentiate to macrophages.

## Disruption of RAS binding to p110α impairs monocyte transendothelial extravasation in response to inflammatory cues

The decrease in classical monocytes observed in the inflammatory abscess of paws from $Pik3ca^{RBD/-}$ mice may indicate a decreased ability to mount an effective immune response. We carried out transwell assays since they are widely used to quantify transendothelial migration. Fibroblasts were seeded in the lower chamber to provide a continuous supply of Ccl2 (*Tsuyada et al., 2012*), as this cytokine is well known for its ability to drive chemotaxis of myeloid cells under inflammatory conditions (*Matsushima et al., 1989*). $Pik3ca^{RBD/-}$ or $Pik3ca^{WT/-}$ BMDMs were seeded in the upper chamber of the transwell. Additionally, we stimulated $Pik3ca^{RBD/-}$ or $Pik3ca^{WT/-}$ BMDMs with LPS and IFN-γ to mimic/recapitulate the pro-inflammatory phenotype typically associated with bacterial infection that causes macrophage activation (*Orecchioni et al., 2019*). Results showed that in both unstimulated and LPS + IFN-γ-stimulated BMDMs, a lower number of $Pik3ca^{RBD/-}$ BMDMs were able to go through the trans-well pore membranes compared to $Pik3ca^{WT/-}$ BMDMs (*Figure 3A*). As expected, when BMDMs were stimulated towards a pro-inflammatory phenotype, a decrease in their migratory ability was observed (*Cui et al., 2018*).

Extravasation entails the migration of monocytes through the endothelium (*Auffray et al., 2009*; *Geissmann et al., 2010*), so we conducted random migration assays of $Pik3ca^{RBD/-}$ and $Pik3ca^{WT/-}$ BMDMs growing in matrigel coated plates under unstimulated conditions or activated towards an inflammatory phenotype by addition of LPS and IFN-γ to the media. Analysis of the data revealed that, under pro-inflammatory conditions, $Pik3ca^{RBD/-}$ BMDMs displayed a decrease in migration speed (*Figure 3B*). Migration was also evaluated in $Pik3ca^{WT/WT}$ BMDMs treated with BYL-719. Results confirmed a decrease in migration speed of macrophages upon treatment with BYL-719 (*Figure 3B*) indicating that inhibition of p110α, either genetically or pharmacologically, reduces the ability of macrophages to migrate under inflammatory conditions.

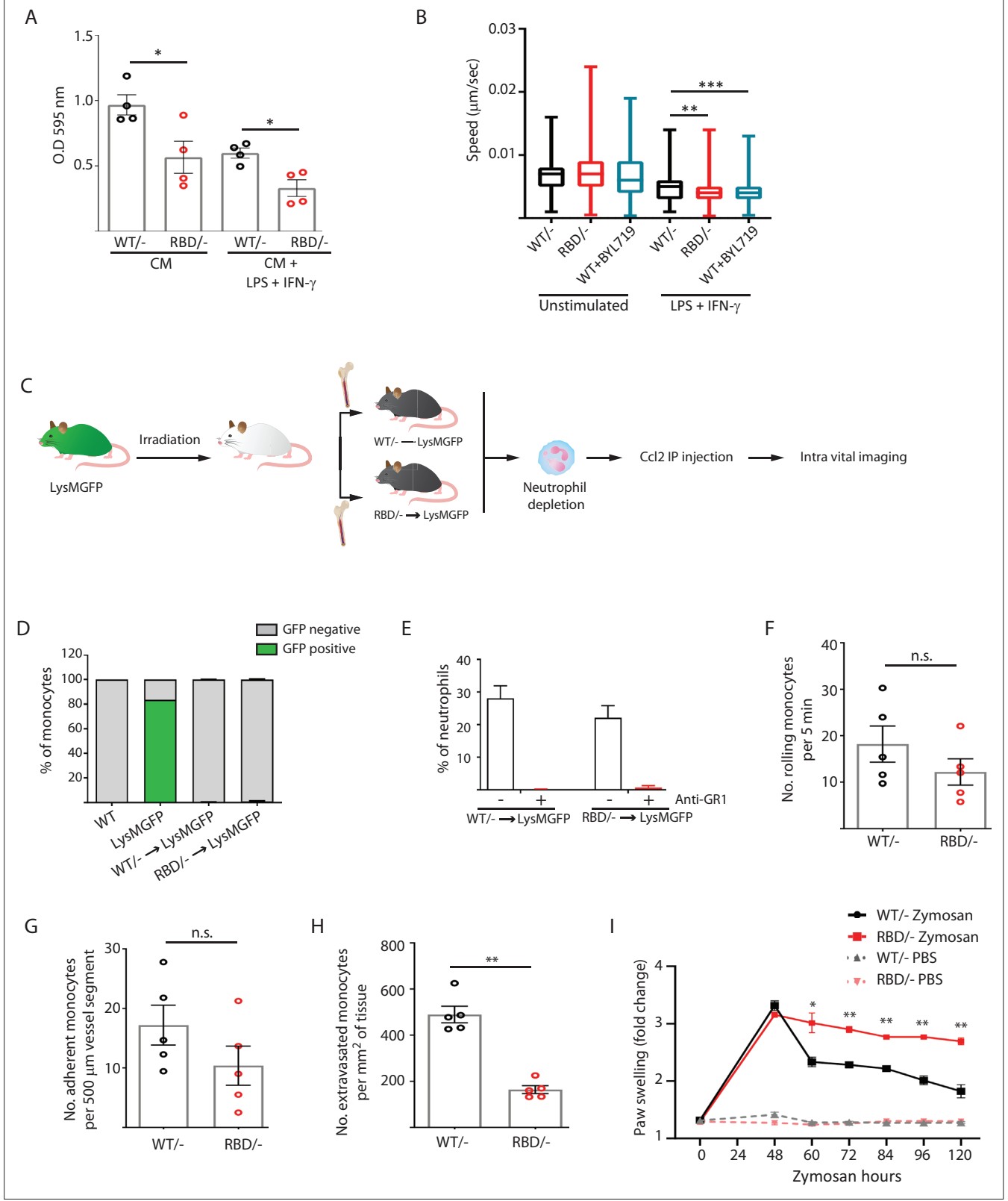

**Figure 3.** Disruption of RAS–p110α interaction impairs transendothelial extravasation to sites of inflammation. (**A**) Graph indicating the quantification of *Pik3ca^RBD/–* and *Pik3ca^RBD/–* bone marrow-derived macrophages (BMDMs) passing through 8 μm membrane pore transwells for 24 hr. Membrane attached macrophages on the lower part of the transwell were stained with crystal violet and quantified. Each dot represents an independent experiment. Error bars indicate mean ± SEM. (**B**) Random migration of *Pik3ca^WT/–*, *Pik3ca^RBD/–*, and *Pik3ca^WT/WT* BMDMs treated with BYL719 (500 ng/ml) was analysed by

*Figure 3 continued on next page*

*Figure 3 continued*

time-lapse video microscopy and cell tracing in the presence or absence of lipopolysaccharide (LPS) (100 ng/ml) and interferon gamma (IFN-γ) (20 ng/ml). (**C**) Schematic representation of myeloid chimera generation strategy. (**D**) Graph showing level of bone marrow reconstitution with *Pik3ca^RBD/–^* and *Pik3ca^WT/–^* myeloid lineage. (**E**) Graph quantifying the number of neutrophils in the blood of chimera mice treated with anti-GR1 (25 μg/mouse/day). Intravital imaging quantification of the number of (**F**) rolling monocytes per 5 min; (**G**) adherent monocytes per 500 μm of vessel segment; (**H**) extravasated monocytes per mm² of tissue. Each dot represents an individual mouse. (**I**) Chimera *Pik3ca^WT/–^* and *Pik3ca^RBD/–^* mice were injected with zymosan or PBS in the back-hind paws and inflammation was measured and plotted over time, *Pik3ca^WT/–^* PBS n = 5; *Pik3ca^RBD/–^* PBS n = 5; *Pik3ca^WT/–^* Zymosan n = 5; *Pik3ca^RBD/–^* Zymosan n = 5. Statistical significance was obtained using Mann–Whitney test (**A–H**) or two-way ANOVA test (**I**): n.s., non-significant; *p < 0.05; **p < 0.01; ***p < 0.001.

The online version of this article includes the following figure supplement(s) for figure 3:

**Figure supplement 1.** *Pik3ca^RBD/–^* monocytes do not extravasate through the endothelium.

To determine if disruption of RAS binding to p110α impairs monocyte ability to extravasate through the endothelium in vivo in response to an inflammatory stress, we analysed monocyte extravasation through the mesenteric vein in response to intraperitoneal Ccl2 injection. It is well established that loss of p110α function leads to significant impairment in endothelial and lymphatic system (*Murillo et al., 2014*; *Soler et al., 2013*), so in order to determine if monocytes from *Pik3ca^RBD/–^* mice presented an alteration in extravasation, we generated chimeric mice in which only the bone marrow were defective in RAS binding to p110α (*Figure 3C*). For this, bone marrow from *Pik3ca^RBD/–^* or *Pik3ca^WT/–^* mice was injected through the tail vein of irradiated Lyz2^GFP^ donor mice (*Faust et al., 2000*). Engraftment of *Pik3ca^RBD/–^* and *Pik3ca^WT/–^* bone marrow could be followed by disappearance of the eGFP signal in donor Lyz2^GFP^ mice (*Figure 3D*, *Figure 3—figure supplement 1A*). Neutrophils (but no monocytes) were depleted using an anti-GR1 antibody (*Figure 3E*, *Figure 3—figure supplement 1B*) to avoid interference with monocyte extravasation. Intraperitoneal injection of Ccl2 was performed in the peritoneum of the chimera mice to induce extravasation of monocytes through the mesenteric vein and rolling, adhesion and extravasation was measured by intravital microscopy. Flow cytometry analysis demonstrated no differences in Ccr2 expression between *Pik3ca^RBD/–^* and *Pik3ca^WT/–^* monocytes (*Figure 3—figure supplement 1C*). We observed that blocking RAS–p110α interaction did not induce a decrease in the number of rolling monocytes (*Figure 3F*) or in the number of monocytes that adhere to the endothelium (*Figure 3G*). However, when extravasation was measured, data showed that disruption of RAS binding to p110α caused a significant decrease in the number of monocytes that were capable of extravasating through the endothelium of the mesenteric vein (*Figure 3H*).

Additionally, in order to determine whether the differences observed in the inflammatory response of *Pik3ca^RBD/–^* mice were due to lack of monocyte extravasation, we repeated the zymosan paw swelling assay in the chimera mice. As previously, we observed that disruption of RAS–p110α only in the immune system led to higher inflammatory response and a delay in the initiation of the resolution phase (*Figure 3I*).

## Disruption of RAS–p110α activation in macrophages induces changes in cytoskeleton reorganization

During transendothelial migration, leukocytes undergo cytoskeletal rearrangements that allow them to squeeze through the tight spaces between endothelial cells and enter the underlying tissue (*Schwartz et al., 2021*; *Vicente-Manzanares and Sánchez-Madrid, 2004*). Therefore, we next sought to determine whether the differences observed in *Pik3ca^RBD/–^* BMDMs extravasation and migration were attributable to altered cytoskeletal dynamics. To do so, we first aimed at examining cell shape and spread in *Pik3ca^RBD/–^* and *Pik3ca^WT/–^* BMDMs in pro-inflammatory conditions after LPS + IFN-γ treatment. Treatment with LPS + IFNγ induced an increase in cell spread in both *Pik3ca^RBD/–^* and *Pik3ca^WT/–^* BMDMs when compared to their respective unstimulated counterparts (*Figure 4A, B*). However, cell spread in LPS + IFNγ activated *Pik3ca^RBD/–^* BMDMs was significantly decreased compared to that observed *Pik3ca^WT/–^* BMDMs (*Figure 4A, B*). Additionally, *Pik3ca^RBD/–^* BMDMs were more elongated and did not acquire the typical rounded shape known to be induced in macrophages after treatment with LPS + IFN-γ (*McWhorter et al., 2013*; *Figure 4A, C*). We also analysed cell height and results showed that *Pik3ca^RBD/–^* BMDMs are higher both in unstimulated conditions and after LPS + IFN-γ stimulation (*Figure 4D*).

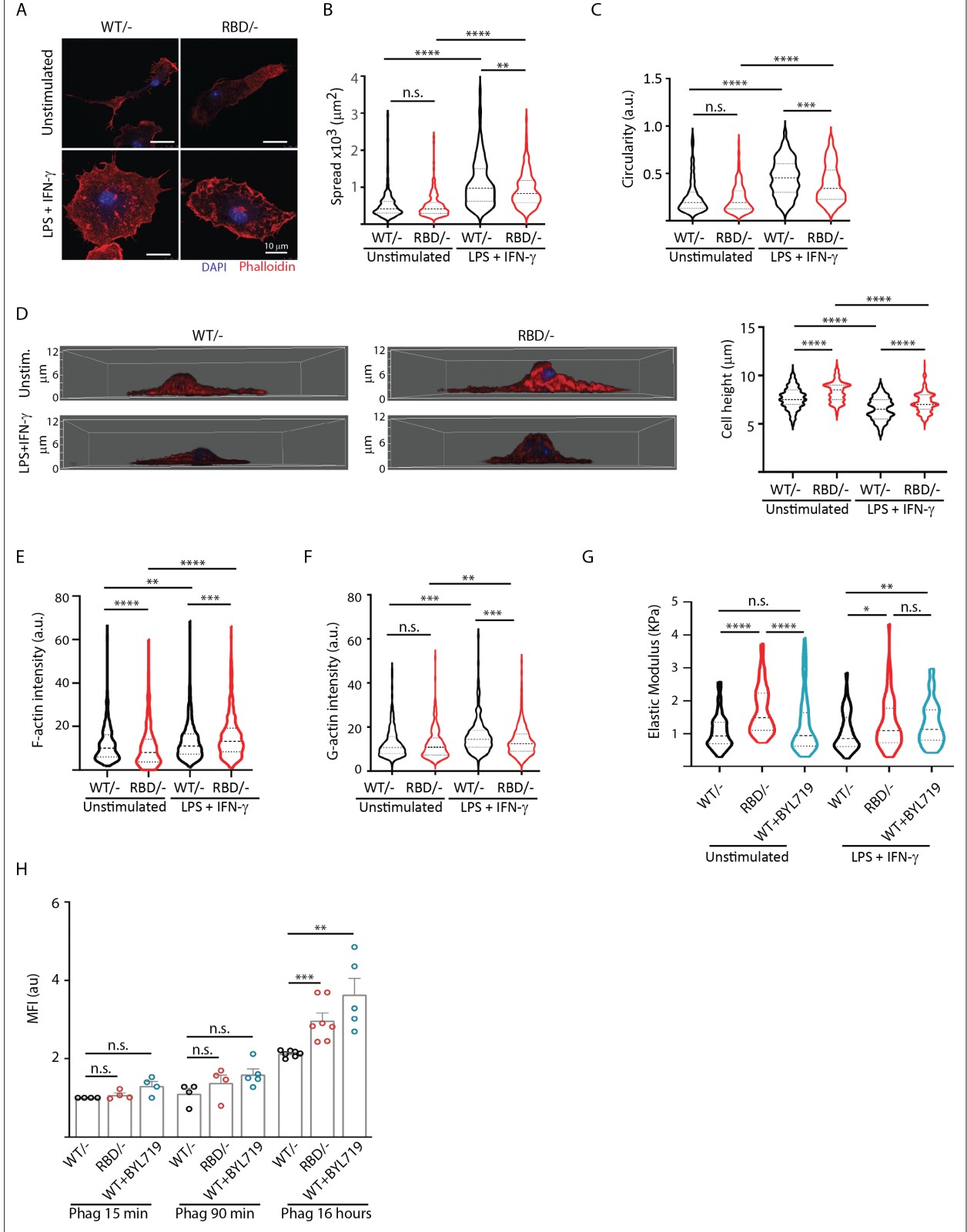

**Figure 4.** Disruption of RAS–p110a signalling impairs bone marrow-derived macrophage (BMDM) ability to remodel their cytoskeleton in response to inflammatory cues. (**A**) Representative images of *Pik3ca*$^{WT/-}$ and *Pik3ca*$^{RBD/-}$ BMDMs unstimulated or activated with lipopolysaccharide (LPS) + interferon gamma (IFN-γ). (**B**) Violin plot quantifying spread area. IF images of *Pik3ca*$^{WT/-}$ and *Pik3ca*$^{RBD/-}$ BMDMs co-stained with phalloidin and DAPI were used to analyse spread area. Results are represented as a violin plot. Three independent biological replicates were analysed (*n* ≥ 250 total cells).

*Figure 4 continued on next page*

*Figure 4 continued*

(**C**) Quantification of cell circularity of $Pik3ca^{WT/-}$ and $Pik3ca^{RBD/-}$ BMDMs unstimulated or activated with LPS + IFN-γ. Quantification was performed using the same images from panel (**A**). (**D**) Representative 3D projections of $Pik3ca^{WT/-}$ and $Pik3ca^{RBD/-}$ BMDMs unstimulated or activated with LPS + IFN-γ and violin plot showing quantification of the corresponding cell height. 3D projections and cell height were analysed using same images used in panel (**A**). (**E**) Violin plot representing F-actin pool in $Pik3ca^{WT/-}$ and $Pik3ca^{RBD/-}$ BMDMs unstimulated or activated with LPS + IFN-γ. Three independent biological replicates were analysed ($n \geq 250$ total cells). (**F**) Violin plot representing G-actin pool (measured by DNAse staining) in $Pik3ca^{WT/-}$ and $Pik3ca^{RBD/-}$ BMDMs unstimulated or activated with LPS + IFN-γ. Three independent biological replicates were analysed ($n \geq 250$ total cells). (**G**) Graph showing stiffness (elastic modulus) of $Pik3ca^{WT/-}$ and $Pik3ca^{RBD/-}$ BMDMs. (**H**) Graph showing phagocytosis of apoptotic cells (efferocytosis) over time in $Pik3ca^{WT/-}$, $Pik3ca^{RBD/-}$, and $Pik3ca^{WT/WT}$ BMDMs treated with BYL719 BMDMs. Statistical significance was obtained using Mann–Whitney test: n.s., non-significant; *$p < 0.05$; **$p < 0.01$; ***$p < 0.001$; ****$p < 0.0001$.

The online version of this article includes the following figure supplement(s) for figure 4:

**Figure supplement 1.** Disruption of RAS–p110α interaction has a differential role in phagocytosis.

Actin is a bona fide regulator of cell shape (*Pollard and Cooper, 2009*), so we next aimed at exploring the actin cytoskeleton in $Pik3ca^{RBD/-}$ and $Pik3ca^{WT/-}$ BMDMs. Actin fractionation assays were carried out in unstimulated or LPS + IFNγ activated $Pik3ca^{RBD/-}$ and $Pik3ca^{WT/-}$ BMDMs. Our results revealed that disruption of RAS–p110α binding caused a decrease in the F-actin pool in unstimulated BMDMs (*Figure 4E*), with no changes observed in the G-actin pool when compared to matched controls (*Figure 4F*). Analysis of actin dynamics after activation with LPS + IFNγ showed that, in pro-inflammatory conditions, the F-actin pool is increased, as expected (*Ronzier et al., 2022*; *Figure 4E*). However, actin polymerization was significantly active in $Pik3ca^{RBD/-}$ BMDMs, leading to a striking increase in F-actin when compared to that observed in controls (*Figure 4E*). In parallel, a decrease in the G-actin pool in $Pik3ca^{RBD/-}$ BMDMs was observed, suggesting an increase in the stabilization of F-actin after disruption of RAS binding to p110α (*Figure 4F*).

We next sought to investigate whether the enhanced F-actin stabilization corresponded to increased cellular rigidity. Consequently, we measured the elastic modulus of $Pik3ca^{RBD/-}$ BMDMs, revealing a significant increase in cell stiffness following RAS–p110α disruption under both basal and pro-inflammatory conditions (*Figure 4G*). To validate that the heightened cell stiffness was attributed to p110α, we treated $Pik3ca^{WT/WT}$ BMDMs with BYL-719 and observed an amplified elastic modulus in these cells when stimulated with LPS + IFN-γ (*Figure 4G*), confirming that the inhibition of p110α indeed results in augmented cellular rigidity.

These observations led us to hypothesize that phagocytosis, which heavily relies on actin rearrangement, might also be affected in $Pik3ca^{RBD/-}$ BMDMs. To test this hypothesis, we assessed the ability of $Pik3ca^{RBD/-}$ and $Pik3ca^{WT/-}$ BMDMs to phagocytose fluorescent microspheres, *B. burgdorferi*, and apoptotic cells. The results revealed varying outcomes depending on the target. When phagocytosing 1 µm non-opsonized green-fluorescent beads, $Pik3ca^{RBD/-}$ BMDMs were less efficient at engulfing the microspheres compared to their WT counterparts (*Figure 4—figure supplement 1A*). Similar results were obtained when $Pik3ca^{WT/WT}$ BMDMs were treated with BYL-719 (*Figure 4—figure supplement 1A*). In contrast, no significant differences were observed between $Pik3ca^{RBD/-}$ and $Pik3ca^{WT/-}$ BMDMs in their ability to phagocytose Borrelia burgdorferi (*Figure 4—figure supplement 1B*).

Lastly, we evaluated the phagocytosis of apoptotic cells by $Pik3ca^{RBD/-}$ BMDMs over extended periods. There were no differences in the initial uptake of apoptotic cells between $Pik3ca^{RBD/}$ and $Pik3ca^{WT/-}$ BMDMs. However, at later time points, $Pik3ca^{RBD/-}$ BMDMs showed an accumulation of phagocytosed particles. Similar results were obtained when $Pik3ca^{WT/WT}$ BMDMs were treated with BYL-719. These data suggest a delay in processing and degradation, indicating a potential role for RAS–p110α in the later stages of phagocytosis (*Figure 4H*).

In summary, our findings suggest that loss of RAS–p110α interaction leads to increased actin polymerization, resulting in stiffer and less deformable cells, which impairs phagocytic efficiency for certain targets. Specifically, RAS–p110α is important for the effective phagocytosis of non-biological particles and may also play a role in the proper processing and degradation of phagocytosed apoptotic cells. The differential effects on various phagocytic targets highlight the complexity of RAS–p110α's role in macrophage biology and underscore the importance of cytoskeletal flexibility in efficient phagocytosis.

## Disruption of RAS–p110α activation impacts the secretome of macrophages

Given our previous observations, we hypothesized that the cytoskeletal alterations observed after disruption of RAS–p110a interaction, may significantly impact the secretory functions of macrophages. The accumulation of apoptotic material in *Pik3ca^{RBD/–}* BMDMs suggests a disruption in normal phagocytic processing, which could influence the release of cytokines and other inflammatory mediators. Thus, we analysed the secretome of *Pik3ca^{RBD/–}* ad *Pik3ca^{WT/–}* BMDMs both under steady-state condition and during phagocytosis. We utilized apoptotic LKR10 cells, a murine lung cancer cell line, as the substrate in our phagocytosis assay. LKR10 cells were exposed to cisplatin for 16 hr and apoptosis was confirmed in a cell viability assay (*Figure 5—figure supplement 1A*). Our goal was to create a more physiologically relevant experimental setting that more closely mimics the complex nature of the inflammatory response. For the secretome analysis, macrophages were incubated with or without apoptotic cells for 16 hr. Culture supernatants were collected, clarified, and subjected to label-free quantitative proteomics analysis.

A total of 127 peptides corresponding to 105 proteins (*Supplementary file 1*) showed differential expression between *Pik3ca^{RBD/}* and *Pik3ca^{WT/–}* BMDM secretomes at steady-state conditions. Additionally, 359 peptides corresponding to 210 proteins (*Supplementary file 2*) were present a significantly different levels in the secretomes of *Pik3ca^{RBD/–}* and *Pik3ca^{WT/–}* BMDM during phagocytosis of apoptotic cells. Next, we compared these peptides with the list of secreted proteins available at The Human Protein Atlas, and removed those that do not correspond to secreted proteins. After this step, 18 proteins were found to be differentially secreted by *Pik3ca^{RBD/–}* and *Pik3ca^{WT/–}* BMDM in steady-state conditions (*Figure 5A*), and 38 by *Pik3ca^{RBD/–}* and *Pik3ca^{WT/–}* BMDM during phagocytosis (*Figure 5B*, *Figure 5—figure supplement 1B, C*). Surprisingly, most proteins were secreted at lower levels in *Pik3ca^{RBD/–}* BMDMs, independently of the conditions under study. Twelve proteins were differentially secreted in both experimental conditions under study (*Figure 5C*).

Functional analysis of differentially secreted proteins in unstimulated BMDMs showed no significant pathways related to these group of proteins (*Figure 5D*). However, differentially secreted proteins in phagocytosing conditions were involved in two main biological processes: complement activation (C1qa, C1qb, and C1qc connected through physical interaction and C3, C9, and Cfb through coexpression) and lysosome function, mainly cathepsins (Ctsd, Ctsb, Ctsz, Cst3, Psap, Anxa1, and Gsn) (*Figure 5E*). Together, the complement cascade and lysosome function work in concert to provide an effective defence against pathogens and promote overall maintenance of cellular homeostasis (*Gordon, 2016*; *Hajishengallis and Lambris, 2016*; *Reis et al., 2019*; *Underhill and Goodridge, 2012*) and data from the secretome analysis of *Pik3ca^{RBD/–}* and *Pik3ca^{WT/–}* BMDMs suggests that RAS activation of p110α may play a crucial role in the regulation of both response pathways. Moreover, these findings align with previous observations showing an accumulation of phagocytosed material in *Pik3ca^{RBD/–}* BMDMs, indicating a potential disruption in the processing and degradation of engulfed targets.

## Disruption of RAS–p110α signalling leads to altered lysosomal function

We next aimed to functionally validate the proteomics data suggesting that RAS–p110α activation regulates lysosomal function in macrophages. First, we performed immunofluorescence analysis of lysosomal-associated membrane protein 1 (LAMP1) to assess lysosomal biogenesis and function in *Pik3ca^{RBD/–}* and *Pik3ca^{WT/–}* BMDMs. Analysis of LAMP1 in *Pik3ca^{RBD/–}* and *Pik3ca^{WT/–}* BMDMs showed a decrease in LAMP1 expression in steady-state conditions (*Figure 6A*) suggesting a decrease in the number of lysosomes after disruption of RAS–p110α interaction. However, *Pik3ca^{RBD/–}* showed increased levels of Lamp1 expression when subjected to phagocytosis of apoptotic cells, suggesting an increase in the number of phagolysosomes present in these cells.

We hypothesized that the increase in Lamp1 expression in BMDMs lacking RAS–p110α interaction during phagocytosis could be attributed to aberrant lysosomal function and phagolysosome retention. Thus, we next evaluated lysosomal activity by using the lysosomotropic dye lysotracker red coupled with epifluorescence analysis, since it is specifically taken up by acidic organelles. As such, its accumulation is proportional to the number of acidic vesicles. Data analysis showed that disruption of RAS–p110α in BMDMs leads to a significant decrease in lysotracker uptake, both in unstimulated conditions and also after activation with LPS + IFN-γ (*Figure 6B*). Similar results were obtained with

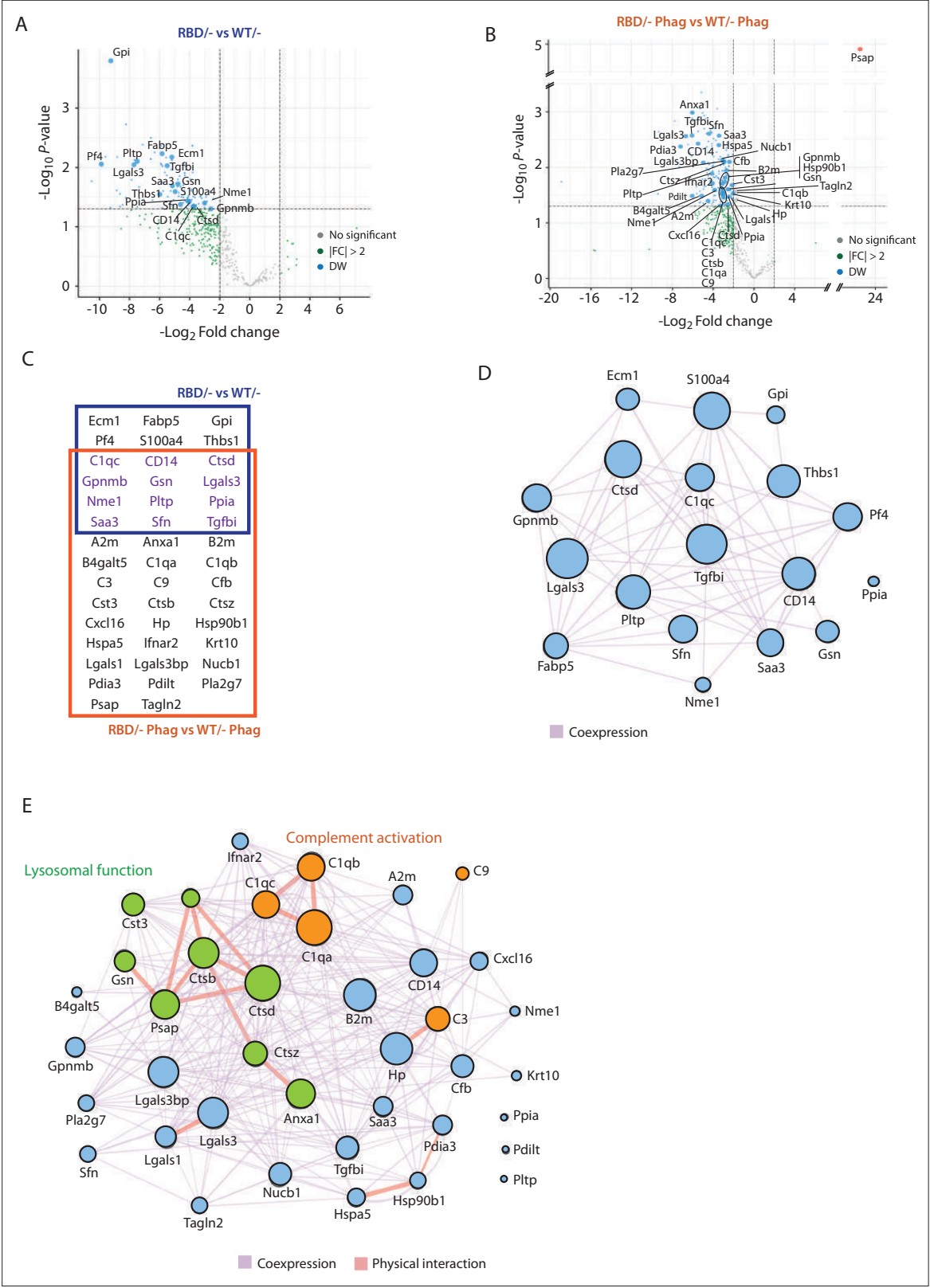

**Figure 5.** Secretome analysis of *Pik3ca^{RBD/−}* bone marrow-derived macrophages (BMDMs) suggested a defect in complement activation and lysosomal function. Volcano plots of secretome analysis from *Pik3ca^{WT/−}* and *Pik3ca^{RBD/−}* BMDMs in non-stimulated conditions (**A**) or phagocytosing apoptotic cells (**B**). The *x*-axis shows the log2 FC of each identified protein and the *y*-axis the corresponding −log10 p value. Statistically significant peptides with FC ≥2 and p-value <0.05 are in blue; peptides that do not pass this threshold are in grey; peptides with FC ≥2 but p-value ≥0.05 are in green. (**C**) Venn

*Figure 5 continued on next page*

Figure 5 continued

diagram showing the overlap of *Pik3ca*[RBD/–] versus *Pik3ca*[WT/–] BMDMs differentially expressed proteins at rest and during phagocytosis of apoptotic cells. Proteins displayed in the blue square represents peptides differentially expressed in resting BMDMs, the brown square represents peptides differentially expressed during phagocytosis, and the peptides differentially expressed in both conditions are displayed in purple. Network analysis of significantly expressed proteins identified in the secretome analysis of *Pik3ca*[RBD/–] versus *Pik3ca*[WT/–] BMDMs in steady-state conditions (**D**) or during phagocytosis of apoptotic cells (**E**). The nodes represent individual proteins, and the edges represent known interactions between proteins, either co-expression (purple) or physical interaction (thick pink). The size of each node reflects the significance of differential expression. Proteins in green have been implicated in lysosomal function, while proteins in orange are members of the complement cascade.

The online version of this article includes the following figure supplement(s) for figure 5:

**Figure supplement 1.** Disruption of RAS–p110α interaction alters the secretome of bone marrow-derived macrophages (BMDMs).

control BMDMs treated with BYL719, the p110α-specific inhibitor. Data analysis showed a decrease in the uptake of lysotracker after p110α inhibition (**Figure 6B**), further suggesting that loss of p110α function in BMDMs results in altered lysosomal pH. To confirm that lysosomal pH of *Pik3ca*[RBD/–] BMDMs was less acidic, we next stained *Pik3ca*[RBD/–] and *Pik3ca*[WT/–] BMDMs with Green lysosensor, a pH-sensitive dye that exhibit a pH-dependent increase in fluorescent intensity upon lysosomal acidification. As shown in **Figure 6C**, *Pik3ca*[RBD/–] BMDMs presented attenuated fluorescent intensity both in unstimulated and activated when compared with that in the control group, indicating that their lysosomal content is less acidic than in *Pik3ca*[WT/–] BMDMs.

Considering that *Pik3ca*[RBD/–] lysosomes were less acidic, we next investigated the expression level and activation of some of the cathepsins identified in the secretome analysis (**Figure 5**) by western blotting. Cathepsins play a critical role in lysosomal protein degradation. They are initially synthesized as inactive precursors and are activated through proteolysis in the lysosome at a low pH (**Yadati et al., 2020**). We found a reduction in the expression and activation of Cathepsins D and B in *Pik3ca*[RBD/–] BMDMs upon stimulation with LPS + IFN-γ (**Figure 6D**). This observation suggests that the impaired lysosomal pH observed in *Pik3ca*[RBD/–] BMDMs could potentially account for the observed decrease in cathepsin activity.

Acidification of lysosomes and cathepsin activation are critical steps for activation of the resolutive stage of the inflammatory response, so we next evaluated the ability of *Pik3ca*[RBD/–] BMDM's lysosomal compartment to degrade internalized particles. We set up another phagocytosis assay in which *Pik3ca*[RBD/–] and *Pik3ca*[WT/–] BMDMs would phagocytose apoptotic cells that were transduced with GFP (which is pH sensitive; **Shinoda et al., 2018**) and also labelled with Celltracker Red CMTPX (pH insensitive). *Pik3ca*[RBD/–] and *Pik3ca*[WT/–] BMDMs were allowed to engulf these apoptotic cells for 16 hr and after this time, apoptotic cells were eliminated by washes and BMDMs were analysed by flow cytometry at different time points to measure GFP signal. Our results showed that GFP signal is lost significantly faster in control BMDMs than in *Pik3ca*[RBD/–] BMDMs (**Figure 6E**). As expected, we did not observe any decay in the red tracker signal. Collectively, these data evidence that *Pik3ca*[RBD/–] BMDMs exhibit a significant impairment in removing engulfed particles, that can be attributed to the absence of lysosome acidification.

We have shown a delayed clearance of apoptotic cells in *Pik3ca*[RBD/–] mice after zymosan injection (**Figure 1**). This, together with previous data took us to analyse the levels of Cathepsin D in the inflamed abscess from the paws of *Pik3ca*[RBD/–] and *Pik3ca*[WT/–] mice. Results showed a significant decrease in the levels of Cathepsin D in the inflamed area of *Pik3ca*[RBD/–] mice (**Figure 6F**) and BYL719-treated mice (**Figure 6G**).

Lysosomal digestion plays a pivotal role in activation of resolutive programs by mediating PPAR activation (**Mota et al., 2021**; **Qiu et al., 2023**). Analysis of PPARδ and PPARγ expression in phagocyting macrophages showed a significant decrease in the expression of PPARδ, but no PPARγ in *Pik3ca*[RBD/–] BMDMs (**Figure 6—figure supplement 1A**), suggesting that resolutive programs might not be activated effectively.

In summary, our findings underscore the crucial role of RAS–p110α signalling axis in maintaining the balance of the inflammatory response and promoting timely resolution, providing further evidence for the significant involvement of RAS–p110α signalling in the response to inflammatory stimuli.

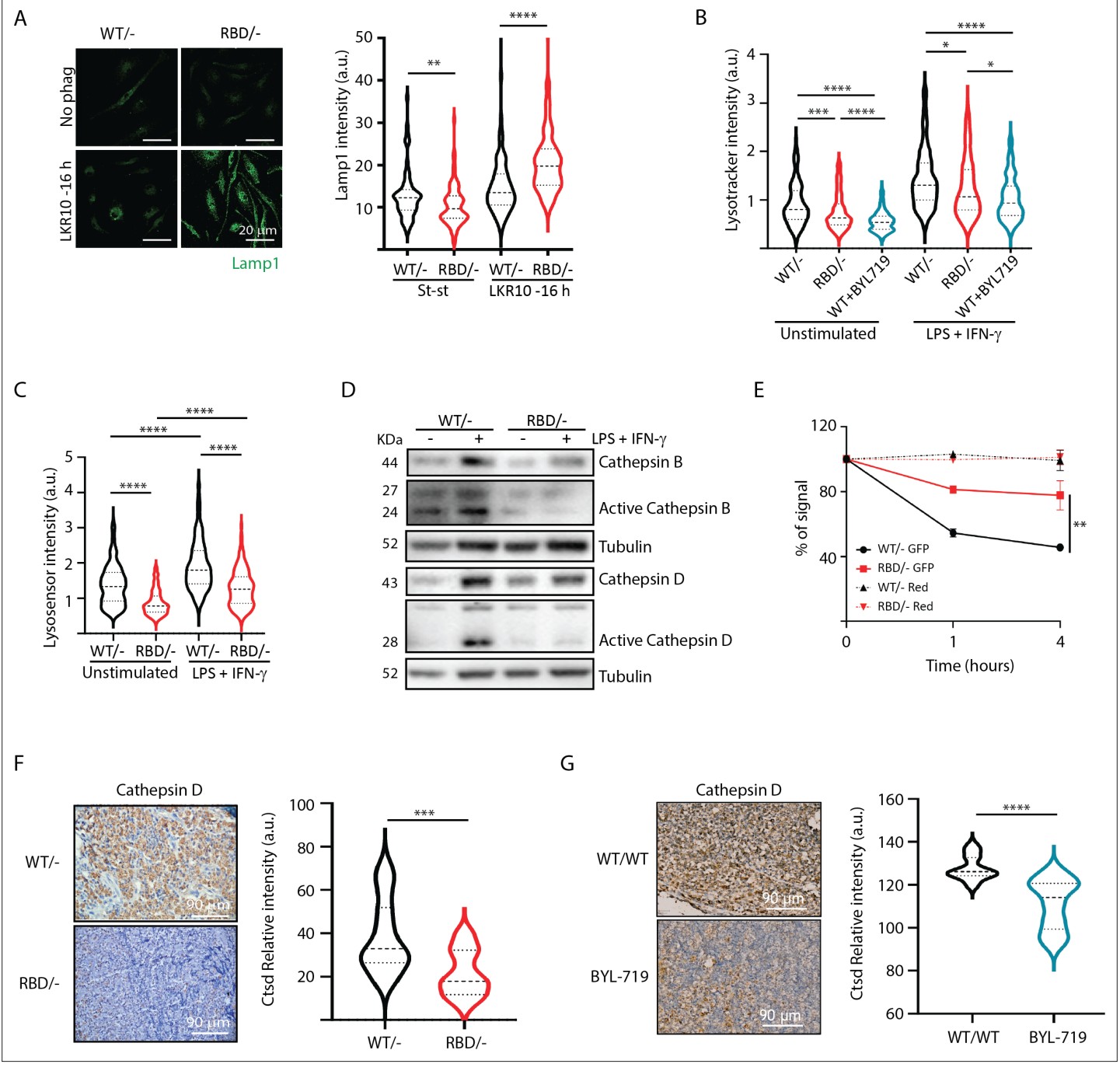

**Figure 6.** Disruption of RAS–p110a activation in bone marrow-derived macrophages (BMDMs) leads to abnormal lysosomal function. (**A**) Representative IF images and quantification analysis of lamp intensity in *Pik3ca^WT/−* and *Pik3ca^RBD/−* BMDMs in steady-state conditions and during phagocytosis of apoptotic LKR10 cells. Three independent biological replicates were analysed (*n* ≥ 250 total cells). Error bars indicate mean ± SEM. (**B**) Quantification analysis of lysosomal function in *Pik3ca^WT/−*, *Pik3ca^RBD/−* and *Pik3ca^WT/WT* BMDMs treated with BYL719 BMDMs unstimulated or activated with lipopolysaccharide (LPS) + interferon gamma (IFN-γ) using Lysotracker staining. Three independent biological replicates were analysed (*n* ≥ 250 total cells). (**C**) Violin plot displaying lysosome pH acidity in unstimulated and LPS + IFN-γ-stimulated *Pik3ca^WT/−* (black), *Pik3ca^RBD/−* (red), and *Pik3ca^WT/WT* BMDMs treated with BYL719 (blue) BMDMs, as determined by Lysosensor staining. Three independent biological replicates were analysed (*n* ≥ 250 total cells). (**D**) *Pik3ca^WT/−* and *Pik3ca^RBD/−* BMDMs were activated with LPS and IFN-γ and expression and activation of Cathepsins B and D were blotted. (**E**) Quantification of phagocytosed apoptotic cells in *Pik3ca^WT/−*, *Pik3ca^RBD/−*, and *Pik3ca^WT/WT* BMDMs treated with BYL719 BMDMs. Apoptotic cells were labelled with a red tracker and allowed to be phagocytosed by control, *Pik3ca^RBD/−* and *Pik3ca^WT/WT* BMDMs treated with BYL719 for 16 hr, measuring the amount of cell tracker that was internalized at different time points. Each dot of the graph indicates a different independent experiment. Error bars indicate mean ± SEM. (**F**) Representative image and quantification of Cathepsin D expression in the inflammatory a bscess of *Pik3ca^WT/−* and *Pik3ca^RBD/−*

*Figure 6 continued on next page*

*Figure 6 continued*

paws injected with zymosan. (**G**) Representative image and quantification of Cathepsin D expression in the inflammatory abscess of paws from control and *Pik3ca^WT/WT^* mice treated with BYL719 injected with zymosan. Statistical significance was obtained using Mann–Whitney test: *p < 0.05; **p < 0.01; ***p < 0.001; ****p < 0.0001.

The online version of this article includes the following source data and figure supplement(s) for figure 6:

**Source data 1.** Membranes corresponding to Cathepsin B, activated Cathepsin B, Cathepsin D, activated Cathepsin D and Tubulin (as loading control) western blots presented in *Figure 6D*, labelled.

**Source data 2.** Membranes corresponding to Cathepsin B, activated Cathepsin B, Cathepsin D, activated Cathepsin D and Tubulin (as loading control) western blots presented in *Figure 6D*.

**Figure supplement 1.** Disruption of RAS–p110α interaction impacts expression of inflammation resolution mediators.

## Discussion

In this study, we provide compelling evidence that disruption of RAS–p110α signalling or chemical inhibition of p110α impairs response to inflammatory stresses due to defects in both monocyte extravasation during the early stages of the inflammatory response and decreased lysosomal function during the later stages (*Figure 7*). Although RAS–p110α signalling disruption affects all the hematopoietic cells, several lines of evidence indicate that macrophages play a central role in the observed phenotype. In our chimera mice experiments, neutrophil depletion did not alter the reduced transendothelial extravasation, suggesting that macrophages are the primary cell type involved. Furthermore, in the paw oedema model, macrophages were the population exhibiting the most significant functional defects. Together, these findings, along with the specific deficiencies observed in myeloid populations, support a predominant role of macrophages in mediating the impaired inflammatory response. Nevertheless, we acknowledge that other myeloid cells, such as dendritic cells or additional immune populations, may also contribute to the global phenotype and further analysis should be performed to address this.

Monocyte extravasation during the inflammatory response is a critical step that allows immune cells to reach the site of infection or injury (*Shi and Pamer, 2011*). Impairment on this process causes slower or inadequate immune response, as macrophages are crucial for detecting and engulfing pathogens or cellular debris at the site of inflammation, as well as delayed resolution of inflammation, resulting in prolonged inflammation and subsequent tissue damage (*Fredman et al., 2012*; *Meizlish et al., 2021*).

Our data show that RAS binding to p110α is involved in macrophage extravasation by modulating actin dynamics. During monocyte extravasation, actin filaments form actin-rich protrusions that are essential for monocyte migration across the endothelium (*Vicente-Manzanares and Sánchez-Madrid, 2004*; *Wilson et al., 2013*). Our data show that, during extravasation, RAS–p110α signalling regulates actin dynamics so monocytes are able to squeeze through endothelial cells (*Vicente-Manzanares and Sánchez-Madrid, 2004*; *Pollard and Cooper, 2009*).

The disruption of RAS–p110α in macrophages emerges as a pivotal determinant in cellular mechanics, elucidating a cascade of events resulting in increased F-actin levels. The increase in cytoskeletal components, particularly F-actin, causes a notable increase in cell stiffness and a concurrent decrease in cell deformability. The orchestration of these changes underscores the intricate balance in cytoskeletal dynamics. Notably, the regulatory role of Rho-GTPases comes into focus as potential mediators of this phenomenon. Rho-GTPases are well-known mediators of actin cytoskeletal rearrangements (*Soriano et al., 2021*), and their dysregulation is known to impact cell mechanics (*Hoon et al., 2016*; *Wolfenson et al., 2019*). The observed increase in cell stiffness and deformability, may thus be governed by the modulation of Rho-GTPase activity. This intricate interplay provides a compelling avenue for further investigation into the molecular mechanisms through which RAS–p110α disruption influences Rho-GTPases, ultimately shaping the biomechanical properties of macrophages and may open avenues for therapeutic interventions targeting cytoskeletal dynamics in macrophages.

Monocyte extravasation shares many features with monocyte egression from the bone marrow. During monocyte egress, monocytes also undergo changes in cytoskeleton dynamics to detach from the sinusoidal endothelial cells and to migrate through the endothelial fenestrae and basement membrane into the bone marrow sinusoids (*Kamnev et al., 2021*). Thus, the decrease in the number of classical monocytes in the *Pik3ca^RBD/–^* mice in blood may be due, at least in part, to a defect in the

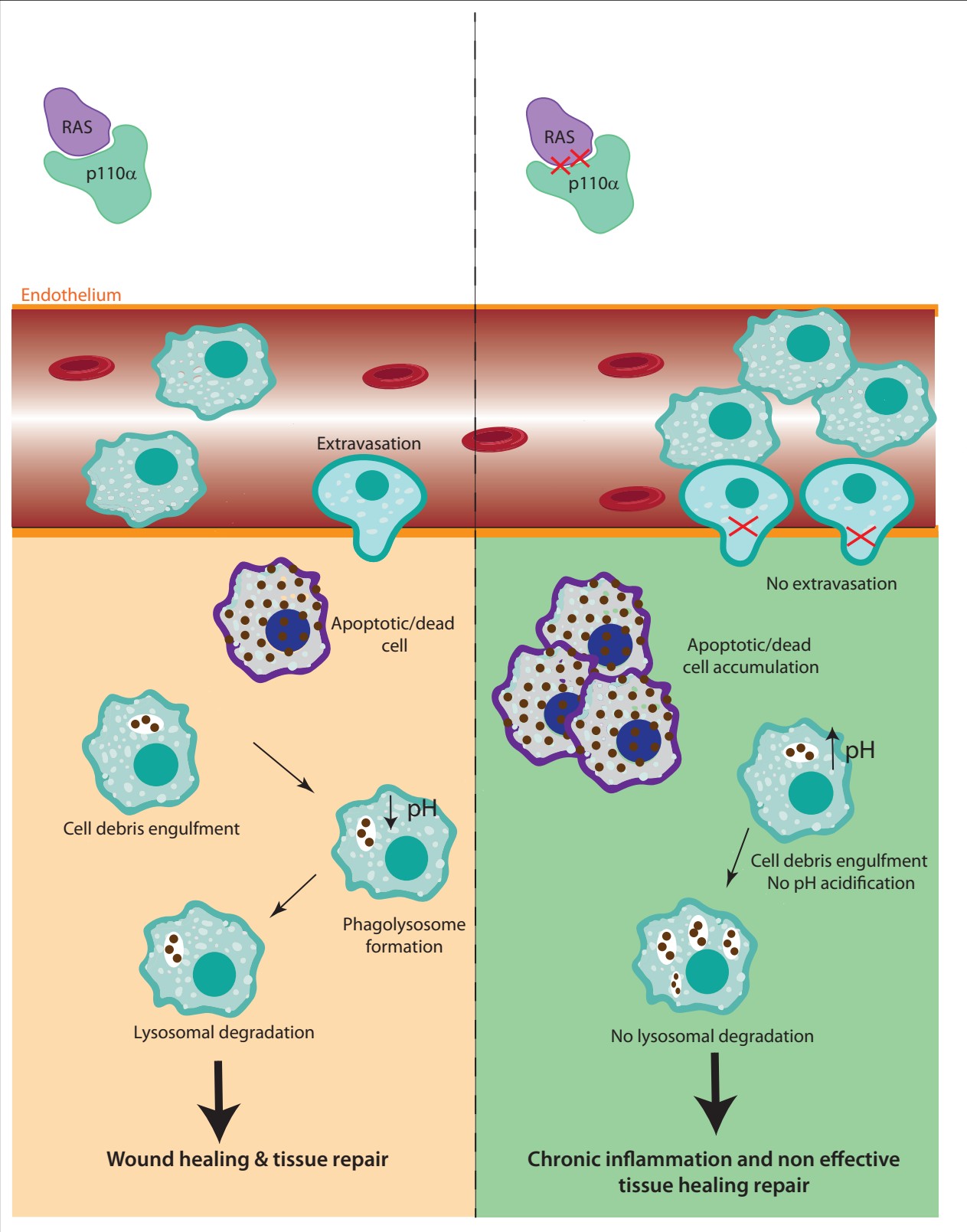

**Figure 7.** The role of RAS–p110α in macrophage-mediated inflammatory responses. In the presence of functional RAS–p110α, monocytes efficiently migrate through the endothelium in response to inflammatory signals. Following extravasation, monocyte-derived and resident macrophages clear apoptotic cells via phagocytosis and lysosomal degradation, facilitating resolution of inflammation. In contrast, loss of RAS–p110α leads to cytoskeletal changes that hinder monocyte transendothelial migration. Additionally, resident macrophages exhibit impaired degradation of phagocytosed particles, resulting in unresolved inflammation and more severe acute inflammatory responses.

extravasation process. This result may also explain, at least partially, the lack of macrophage recruitment to lung tumours found in previous studies using the RBD mouse model (*Castellano et al., 2013*; *Murillo et al., 2014*).

Our findings also revealed a crucial role of RAS–p110α activation in the acute phase of the inflammatory response by regulating effective lysosomal degradation of phagocytosed and engulfed material. Phagocytosis constitutes a vital step in the inflammatory response, whereby phagosomes bind to lysosomes to form phagolysosomes in which pathogens are eradicated to facilitate an appropriate host response. This response encompasses antigen presentation to engage T-cell responses, secretion of inflammatory mediators that guide the adaptive immune response, and initiation of tissue repair mechanisms (*Gordon, 2016*; *Underhill and Goodridge, 2012*). Our data provide evidence that the activation of RAS–p110α signalling pathway is involved in the critical process of lysosomal acidification, which is essential for the efficient degradation of internalized particles and the activation of proteolytic enzymes, ultimately resulting in the formation of fully functional lysosomes. Consequently, lack of lysosome acidification impairs the expression and activation of important proteases such as Cathepsins B and D. The proper functioning of lysosomes is essential for mounting a robust response to inflammatory stress. Lysosomes play a central role in the breakdown of pathogens and dead cells by providing the necessary degradative enzymes and maintaining an acidic environment that facilitates the degradation of engulfed particles (*Ballabio and Bonifacino, 2020*; *Saftig and Puertollano, 2021*). When lysosomal function is compromised in macrophages, the degradation of phagocytosed material becomes impaired, leading to the accumulation of toxic debris. This accumulation subsequently triggers inflammation and causes damage to the surrounding tissues, leading to chronic inflammation and appearance of lysosomal storage diseases (*Scerra et al., 2022*; *Ferreira and Gahl, 2017*).

Our data showed that the disruption of Ras–p110α in macrophages presents a multifaceted impact in inflammatory response, notably manifesting as a reduction in NF-κB activation and cathepsin expression coupled with diminished lysosomal function. NF-κB, a pivotal regulator of inflammatory responses (*Liu et al., 2017*; *Tak and Firestein, 2001*), is intricately linked to lysosomal dynamics, with lysosomes playing a crucial role in modulating NF-κB signalling through the degradation of relevant molecules (*de Mingo et al., 2016*; *Yang et al., 2015*). Additionally, the observed decrease in NF-κB activation aligns with the decline in lysosomal function, which has previously been suggested in previous reports (*Chang et al., 2013*; *Liu et al., 2003*). This bidirectional influence, where perturbations in lysosomal function coincide with alterations in NF-κB activity, underscores the intricate and context-dependent nature of this regulatory network, suggesting a plausible mechanistic interdependence. Our data suggest that disruption in Ras–p110α signalling cascade could trigger downstream effects impacting both NF-κB activity and lysosomal integrity. This nexus between Ras–p110α signalling, NF-κB activation, cathepsin expression, and lysosomal function underscores the complexity of cellular regulatory networks. Further analysis of the molecular pathways connecting these observations holds the potential to reveal novel insights into the coordinated regulation of inflammatory and lysosomal processes in macrophages. Thus, further investigations are necessary to decipher the causal relationships and unravel the broader implications of Ras–p110α disruption in shaping these intricate cellular responses during the inflammatory response.

The possible functional link between the increased actin polymerization and lysosomal dysfunction observed in *Pik3ca*^RBD/-^ mice remains an unanswered question. Lysosome acidification and actin polymerization are tightly interconnected processes that have pivotal roles in numerous cellular functions (*Kast and Dominguez, 2017*; *Taunton et al., 2000*; *Ettelt et al., 2023*). Studies have demonstrated that lysosome acidification can influence actin polymerization dynamics through the activation of specific actin-binding proteins and the modulation of actin-regulatory proteins (*Kast and Dominguez, 2017*; *Stockinger et al., 2006*). Conversely, disruption in lysosome acidification, such as impaired proton pump activity or lysosomal storage disorders, have been associated with changes in actin polymerization and the organization of the cytoskeleton. Notably, it has been reported that depolimeryzation of F-actin plays a crucial role in assembling the macromolecular components of the acidification machinery in nascent endosomes (*Hryciw et al., 2004*). Therefore, the intricate relationship between lysosome acidification and actin polymerization suggests a potential reciprocal influence, where perturbations in one process could impact the other. Further investigation is required to determine the precise regulatory mechanisms by which p110α influences both lysosome acidification and actin polymerization, whether it occurs in a linear manner or through separate pathways.

This study provides valuable insights into the mechanisms that govern the immune response to inflammation, particularly emphasizing the essential role of RAS–p110α signalling. Our results underscore the pivotal role of RAS–p110α signalling in both the initiation and resolution of inflammation. The impaired ability of RAS–PI3K-deficient monocytes and macrophages to effectively migrate, initiate a robust inflammatory response, and resolve inflammation highlights the critical importance of this pathway in maintaining immune homeostasis. The persistent inflammation observed in these models may contribute to the development and perpetuation of chronic inflammatory conditions. Given the centrality of p110α in both initiating and resolving inflammation, its dysfunction could be a key factor in the chronic, dysregulated inflammation characteristic of diseases such as rheumatoid arthritis, inflammatory bowel disease, psoriasis, or systemic lupus erythematosus.

The potential of p110α as a therapeutic target in these conditions is especially intriguing. The recent identification of a p110α small molecule activator (*Gong et al., 2023*) offers a promising tool to explore this avenue. By transiently enhancing p110α function, it may be possible to promote the resolution of inflammation and thereby alleviate the symptoms and progression of these inflammatory diseases. Further research into this therapeutic strategy could open new pathways for treating chronic inflammatory disorders, providing much-needed relief for patients affected by these debilitating conditions.

## Methods

### Animal studies

*Pik3ca*$^{RBD/lox}$/Rosa26$^{CreERT2/WT}$/KRAS$^{G12D/WT}$ mouse model was kindly given by Dr. Downward Laboratory (*Castellano et al., 2013*; *Gupta et al., 2007*).

For removal of *Pik3ca*- floxed allele, *Pik3ca*$^{RBD/lox}$ and *Pik3ca*$^{WT/flox}$ mice (*Yang et al., 2015*) were given 3.2 mg tamoxifen (Sigma) dissolved in 80 μl of corn oil by oral gavage once per day during 3 consecutive days. Efficiency of tamoxifen treatment was routinely performed by genotyping for the presence of the floxed allele.

For **paw oedema studies** *Pik3ca*$^{RBD/–}$ and *Pik3ca*$^{WT/–}$ mice (*Yang et al., 2015*) were divided into groups of four 2 weeks before oedema induction. Before inducing the paw oedema, the mice were anesthetized with 4% isoflurane. To induce the oedema, mice received ipsilateral i.pl. injection (30 μl) of either zymosan (10 μg/μl, Sigma-Aldrich) or PBS into the back-hind paw. Injection of 0.1 mg/kg Buprenorphine (NOAH, Vetergesic) was given for pain prevention. Paw thickness was measured using a calliper every hour during the first 6 hr after injection and then at 8, 10 hr and twice per day afterwards. Buprenorphine was injected twice per day during the length of the experiment.

Animals were randomly assigned to experimental groups to ensure unbiased allocation and minimize potential confounding factors. Animals were randomly assigned to groups with similar numbers, ensuring all were born on the same date to control for age-related factors. No formal sample size calculation was conducted. Blinding was implemented during data collection and analysis to reduce bias, with researchers unaware of the group allocations when assessing outcomes. Inclusion and exclusion criteria were pre-established to ensure consistency; animals showing pre-existing health conditions were excluded from the study to maintain experimental integrity; only females were included in this study. Mice were kept, managed, and sacrificed in the NUCLEUS animal facility of the University of Salamanca according to current European (2007/526/CE) and Spanish (RD 1201/2005 and RD53/2013) legislation. All experiments were approved by the Bioethics Committee of the Cancer Research Center. All animal procedures were conducted in accordance with the guidelines and humane endpoints established in our animal experimental license to minimize pain, suffering, and distress. Interventions such as Buprenorphine were employed as needed, and procedures were designed to be minimally invasive. Animals were monitored daily for expected and unexpected signs of pain or distress, with specific criteria—such as weight loss, lethargy, abnormal grooming—set as humane endpoints to ensure early intervention if required. No unexpected adverse events occurred during the study.

### Isolation, culture, and treatments of BMDM

Bone marrow cells from tibias and femurs of 12- to 14-week-old *Pik3ca*$^{RBD/Lox}$ mice and *Pik3ca*$^{WT/Lox}$ littermates were cultured with DMEM supplemented with 10% FBS, 100 units/ml penicillin, 100 μg/

ml streptomycin, 2 mM L-glutamine and 20 ng/ml M-CSF for 7 days. 4-hydroxytamoxifen (Sigma-Aldrich) (100 nM) was added to culture media on day 3 to eliminate *Pik3ca*-Lox allele. The differentiated BMDM were then detached using cell dissociation buffer (C5914-100, Merck) and cultured in DMEM supplemented with 10% FBS, 100 µg/ml streptomycin, 2 mM L-glutamine, 20 ng/ml M-CSF for unstimulated BMDMs. For macrophage polarization towards an inflammatory phenotype 20 ng/ml IFNγ (Peprotech) and 100 ng/ml LPS (Sigma-Aldrich) was added to culture media.

## Generation of chimeric animals
Chimeric mice exhibiting WT or RBD-deficient leukocytes were generated by lethal irradiation with 5.5 Gy twice, 4 hr apart of Lyz2[GFP] recipient animals (mice exhibiting endogenously GFP fluorescent monocytes and neutrophils) followed by an injection of bone marrow cells (1.5 × 10[6] cells/recipient i.v.) from C57BL/6 WT or RBD donor mice. Chimerism was then assessed 4 weeks later by flow cytometry from blood samples (reconstitution of 99.8 ± 0.2% and 99.8 ± 0.1% for WT and RBD-deficient donor cells, respectively; *n* = 5 mice per group). Lyz2[GFP] donors were kindly given by Dr. Voisin's laboratory.

## Neutrophil depletion
Neutrophil depletion of chimeric mice was induced by intraperitoneal injection of anti-GR1 25 µg/mouse/day for 3 days. Numbers of blood circulating monocytes and neutrophils were quantified by flow cytometry pre- and post-depletion. Neutrophils were found to be reduced by 99.5%, while this anti GR1-depleting protocol had no effect on blood monocyte proportion (*n* = 5 mice/group).

## Brightfield intravital confocal microscopy
Mesenteric inflammation was induced following intraperitoneal injection of mouse recombinant CCL2 (500 ng/mouse in 500 µl of PBS). Six hours later, anesthetized chimeric mice (150 mg/kg ketamine, 7.5 mg/kg xylazine, i.p.) were placed in supine position on a heating pad (37°C) for maintenance of body temperature. The mesenteric vascular bed was exteriorized, placed on a purpose-built stage of an upright brightfield microscope (Zeiss Axioskop). Mesenteries were superfused with warmed (37°C) Tyrode's solution (Sigma). After a 5-min equilibration period, analysis of leukocyte–endothelium interactions was made in at least 9 (and up to 16) randomly selected segments (100 µm in length) of post-capillary venules (20–40 µm in diameter) for each mouse. Leukocyte rolling was quantified by counting the number of rolling cells passing a fixed transversal line in the middle of the vessel segment for 5 min. Leukocyte adhesion (stationary position of the cell for 30 s or longer) was quantified along a 100-µm vessel length and data were normalized as the number of cells per 500 µm vessel segments. Leukocyte extravasation response was quantified within 50 µm on either side of the 100 µm vessel segment in the perivenular tissue; and data were normalized as the number of extravasated leukocytes per mm2 of extravascular tissue. At the end of the analysis period, mice were humanely killed by cervical dislocation.

## BMDM elastic modulus
The elastic modulus of BMDMs was quantitatively assessed utilizing atomic force microscopy (AFM). Specifically, cells were cultured on cover glass slides, which were subsequently positioned in a specialized BIO-AFM setup integrated with an inverted optical microscope (Nikon TE2000). The BIO-AFM was equipped with a V-shaped, four-sided pyramidal silicon nitride tip (Bruker AFM Probes) to facilitate accurate force measurements. To avoid localized variations and ensure data representativeness, no more than three cells were selected from a single field of view for mechanical characterization, and these cells were never contiguous. Subsequently, the stiffness of each individual cell was characterized through measurements obtained at three perinuclear points. For each point, force–displacement (*F* vs. *z*) curves were captured (10 µm amplitude at a speed of 5 µm/s). The determination of the elastic modulus was performed based on the analysis of force–displacement curves. A total of five curves were recorded for each perinuclear point at an indentation depth of 300 nm. Data were analysed by fitting the curves to the Hertz model as previously described (*Jorba et al., 2017*; *Rico et al., 2005*). Finally, the elastic modulus for each cellular population under distinct experimental conditions was calculated based on a minimum of fifteen measurements per independent cell, with each condition comprising at least 10 independent cells (*n* ≥ 10).

## Cytokine arrays

BMDMs were cultured in 6-well plates at a density of $1 \times 10^6$ cells per well. Cytokine production was analysed in unstimulated and M1 macrophages using commercial mouse cytokine array (R&D Systems, #ARY006). Cells were lysed at 4°C for 30 min using a lysis buffer containing 1% Igepal CA-630, 20 mM Tris-HCl (pH 8.0), 137 mM NaCl, 10% glycerol, 2 mM EDTA, 10 µg/ml Aprotinin, 10 µg/ml Leupeptin, and 10 µg/ml Pepstatin, following the manufacturer's protocol. Lysates were then centrifuged at 16,100 RCF for 15 min at 4°C. The supernatant was collected, and protein concentration was determined using a Qbit 2.0 Fluorometer. A total of 1400 µg of protein was diluted to a final volume of 1 ml with Array Buffer 6, followed by the addition of 0.5 ml Array Buffer 4 and 15 µl of the reconstituted Mouse Cytokine Array Panel A Detection Antibody Cocktail. The samples were mixed and incubated for 1 hr on a shaker at room temperature. Nitrocellulose membranes from the mouse cytokine array were blocked with Array Buffer 6 for 1 hr at room temperature on a shaker. The blocking buffer was then removed, and the sample mixture was added to the arrays, which were incubated overnight at 4°C on a shaker. Membranes were washed with Wash Buffer provided by the manufacturer. Membranes were then incubated with 1× Streptavidin-HRP in Array Buffer 6 for 30 min at room temperature. After further washing, the Chemi Reagent Mix was applied, and signals were detected using an Amersham Imager 600 (GE Healthcare, #29-0981-07 AC).

## Immunofluorescence

BMDM were fixed using 4% paraformaldehyde, permeabilized with 0.1% Triton X-100 and blocked for 1 hr with 3% BSA in PBS before incubation with the primary antibodies, used at a 1:100 dilution: Lamp1 (#553792, BdPharmigen), Deoxyribonuclease I-Alexa Fluor 488 Conjugate (#D12371, Invitrogen, 1:2000). To stain actin cytoskeleton, Alexa Fluor 647 Phalloidin (Invitrogen, 1:10,000) was directly added to the primary antibody mixture. Alexa Fluor 488- or Alexa Fluor 555-conjugated secondary antibodies (Invitrogen) were used to detect the indicated proteins at a 1:1000 dilution. Cells were counterstained with DAPI on the mounting solution (ProLong Gold Antifade Reagent with DAPI, Invitrogen). Images were taken using a Zeiss LSM510 confocal microscope or Leica DM6 B THUNDER Imager 3D Tissue.

## Transwell migration assay

Transwell migration assays were carried out using the 6.5 mm Transwell with 8.0 µm Pore Polyester Membrane Insert (Corning). $9 \times 10^4$ MEFs from wild-type mice were used as a chemo-attractant to encourage macrophage migration. $8 \times 10^5$ BMDM were seeded in the transwell. Transwells were performed following the manufacturer's instructions.

## Random migration assay

For random migration assays, BMDMs were seeded in 24-well plates coated with matrigel (0.5 mg/ml) and labelled using CellTracker Red CMTPX Dye 1 µM (Thermo Fisher) for 30 min. Twenty-four hours later LPS + IFN-γ was added when necessary. Triplicates of each condition and genotype were prepared. Time-lapse imaging was carried out for 24 hr. One image was taken every 10 min within the same well using a Nikon microscope driven by Metamorph (Molecular Devices, Chicago, IL, USA). A total of 80–100 cells per condition were tracked using the Fiji plugin Trackmate. Pre-processing was done using Mexican hat filter 3.0 radius to increase particle detection. Images were segmented using the fluorescence channel with the Laplacian of the Gaussian detector with a 30-µm estimated particle diameter, a 10.0 threshold and median filter option selected. Segmented objects were linked from frame to frame with a Linear Assigment Problem (LAP) tracker with 45 µm frame-to-frame linking distance and 2 frame gap closure. Criteria for track acceptance were track duration at least the 90% of the video. Tracks were visually inspected for completeness and accuracy of the tracking.

## Secretome mass spectrometry

Samples for secretome analysis were prepared as previously described (*Álvarez-Teijeiro et al., 2018*). In brief, 100 µg of proteins were digested into peptides using trypsin and peptides were desalted using Oasis HLB extraction cartridges (Waters UK Ltd) and eluted with 50% acetonitrile (ACN) in 0.1% trifluoroacetic acid (TFA).

Dried peptides were dissolved in 0.1% TFA and analysed by nano ACQUITY liquid chromatography (Waters Corp, Milford, MA, USA) coupled on-line to a tandem LTQ Orbitrap XL, mass spectrometer (Thermo Fisher Scientific) (*Casado et al., 2013*). Gradient elution was from 5% to 25% buffer B in 180 min at a flow rate 300 nl/min with buffer A being used to balance the mobile phase (buffer A was 0.1% formic acid in water and B was 0.1% formic acid in ACN). The mass spectrometer was controlled by Xcalibur software and operated in the positive mode. The spray voltage was 1.95 kV and the capillary temperature was set to 200°C. The LTQ Orbitrap XL was operated in data-dependent mode with one survey MS scan followed by 5 MS/MS scans. Label-free quantitative proteomics analysis was performed using three independent biological samples per group. Additionally, each sample was analysed in technical duplicates. To ensure robust quantitative analysis, we utilized LTQ Orbitrap XL tandem mass spectrometry (MS/MS) to generate six distinct mass spectral profiles from each group.

MS raw files were converted into Mascot Generic Format using Mascot Distiller (version 2.3.0) and searched against the SwissProt database (release December 2015) restricted to human entries using the Mascot search daemon (version 2.3.1). Allowed mass windows were 10 ppm and 600 mmu for parent and fragment mass to charge values, respectively. Variable modifications included in searches were oxidation of methionine, pyro-glu (N-term) and phosphorylation of serine, threonine, and tyrosine.

Spectral counting quantification method relies on the number of times peptides are identified by tandem mass spectrometry (with expectancy value <0.05) from a given protein. Spectral counts were obtained from Mascot result (DAT) files using a python script written in house in the Mascot Parser Toolkit environment (version 2.4.x).

The mass spectrometry proteomics data have been deposited to the ProteomeXchange Consortium via the PRIDE (*Perez-Riverol et al., 2022*) partner repository with the dataset identifier PXD057794 and 10.6019/PXD057794.

## Proteomic data analysis

The proteomic data obtained consisted of 6844 peptides and 30 samples: *Pik3ca*[WT/−] and *Pik3ca*[RBD/−] BMDMs in steady-state conditions (labelled as cell samples: WT/- and RBD/-), phagocytosing apoptotic cells (labelled as cell samples: WT/-Phag and RBD/-Phag), and the apoptotic LKR10 cells alone (labelled as LKR). For each of these samples, proteomic experiments were performed with six replicates (three biological replicates × two technical replicates), yielding a dataset of 30 samples.

The first analytical step was to remove all peptides for which there was no information contained in the proteomic raw data matrix and the peptides for which 85% or more of the signal values were missing. All these peptides were specific of mouse proteins and in many cases were unique. The corresponding proteins were annotated and labelled together with each measured peptide. Next, low-quality samples were also removed, testing the overall signal per sample to identify if there were clear outliers with very low signal or with a very different signal distribution. Comparison of the overall signal distributions of the 30 samples (comparing boxplots) and identified 3 samples that were very different were obtained and discarded (WT/-Phag_s3r2 (sample 3, replicate 2), RBD/-_s2r1 and LKRc_s1r1). These three samples showed a median signal in their distributions that deviated >20% from the median signal of the distributions of all other samples.

Differential expression analysis for each peptide of each protein was next performed. The algorithm used to carry out this analysis was *limmaVoom* within EdgeR R package (*McCarthy et al., 2012*; *Robinson et al., 2010*). Prior to this analysis, a Bartlett test was performed to see the homogeneity of variances, verifying that for this data we cannot consider equality of variances and this factor was included in the differential expression algorithm. With this algorithm, normalization factors to use a posteriori were calculated and, transformation and calculation of the variance weights was performed. The model to fit before using *Voom* as specified since it uses the variances of the model's residuals (observed − fitted). Finally, an estimation of the contrast for each feature tested (i.e. each peptide) was carried out using the *Empirical Bayes* approach in *limma* as previously described (*Campos-Laborie et al., 2019*). Peptides were ordered by the p.value of the *limma* test considering significant peptides changed only with a p.value below 0.05 and with a log2(FoldChange) >|2|. All these analyses were performed using the statistical computing language R and packages or libraries obtained from R-cran (https://cran.r-project.org/) or Bioconductor (https://www.bioconductor.org/).

Cytoscape software (v3.9) (*Killcoyne et al., 2009*) including GeneMania app (*Franz et al., 2018*) was then used to generate and visualize protein–protein networks of the significantly altered proteins selected in secretome analysis of unstimulated and phagocyting BMDMs. This tool provides information on protein–protein associations based in co-expression studies and also based in physical interaction studies.

## Western blot analysis

Immunoblot was performed per a general western blot protocol (Abcam). Total protein was extracted using Cell Lysis Buffer (Cell Signaling Technology) supplemented with c0mplete mini protease inhibitor cocktail (Roche), 50 mM sodium fluoride and 1 mM of PMSF. Protein was quantified using Bradford Method (Bio-Rad). 20 µg of protein was separated by SDS–PAGE and transferred to 0.2 um pore-size PVDF membranes (Sigma-Aldrich). Blots were probed using the following antibodies, at a concentration 1:1000 unless otherwise stated: cathepsin B (12216-1-AP, Proteintech), cathepsin D (21327-1-AP, Proteintech), and α-tubulin (ab15246, Abcam; concentration 1:5000). Horseradish peroxidase-conjugated secondary antibodies (Amersham) were used (1:5000) and detected using an enhanced chemiluminescent substrate (Amersham). Signal was detected using an iBright 1500 System (Invitrogen).

## Flow cytometry analysis

Single-cell suspensions from cultured cell, spleen or blood monocytes were generated from mice, washed twice in staining buffer and incubated with 1:100 Fc-block (BD Biosciences, #553142) diluted in FACS buffer. Cells were subjected to surface antibody staining with labelled antibodies diluted in staining buffer for 30 min at 4°C: CD3-PE-Cy7 (#100328, Biolegend), CD4-BV605 (#100548, Biolegend), CD8-APC (#100712, Biolegend), CD19-PerCP-Cy5.5 (#115534, Biolegend), CD45-BV785 (#103149, Biolegend), Ly6C-PerCP-Cy5.5 (#128012, Biolegend), Ly6C-E450 (#48-5932-82, eBioscience), Ly6G-AF700 (#56-5931-82, eBioscience), CD11b-BV650 (#101239, Biolegend), F4/80-PE-Cy7 (#123114, Biolegend), F4/80-PE (#123110, Biolegend), and CCR2 (CD192)-PE-Vio 770 (#130-108-724, Miltenyi Biotec). After incubation, cells were washed in staining buffer and analysed immediately. For all staining, isotype controls were used.

The gating strategy for analysing distinct cellular populations began by isolating cells positive for the CD45 marker. Subsequently, B cells, identified by their positivity for CD19 and negativity for CD11b, were selected. To determine the frequency of T cells, the CD45+ cell population, devoid of both CD19 and CD11b, underwent examination for CD3 expression. CD3+ cells were further categorized based on CD4 and CD8 expression, distinguishing CD4+, CD8+, and double-negative (DN) T cells. Following the application of these gating criteria across all samples, the percentages of CD45-positive cells expressing CD19 and T cells (CD3+) within the CD45+ population were calculated. Additionally, myeloid populations were analysed by quantifying the proportion of CD11b+ cells. To distinguish between circulating monocytes and granulocytes, we assessed their expression of Ly6G (granulocytes) and Ly6C proteins (monocytes). Classical monocytes were characterized by negativity for Ly6G and positivity for Ly6C, while alternative monocytes lacked both markers, and neutrophils/granulocytes displayed weak positivity for Ly6C and positivity for Ly6G.

Samples were acquired on a BD LSR FORTESA FACS or FACS Aria III machine that uses FACS DIVA software (BD Biosciences). Compensation was performed using 1 drop of Ultracomp ebeads (eBioscience) in 300 µl of FACS buffer. 1 µl of each antibody used in the pool was mixed with 100 µl of compensation beads solution and acquired. A total of 50,000 cells per mouse were analysed. Analysis was performed with FlowJo software (FlowJo V10.4). Once the different pools were compensated samples were acquired.

## Phagocytosis assay

For phagocytosis assay, BMDMs in suspension in DMEM without FBS were labelled with 1:200 red cell tracker (Molecular Probes) for 30 min at 37°C. Cells were then centrifuged for 5 min at 300 × *g* at 4°C, supernatant was removed and labelled BMDMs were washed twice with 5 ml of PBS and plated overnight in complete DMEM containing 20 ng/ml M-CSF. On the following day, 100 ng/ml LPS (Sigma-Aldrich) was added to BMDMs overnight.

LKR10 cells (murine lung cancer cell line) were stained with red cell tracker (Molecular Probes) as previously described for macrophages. Stained cells were plated in complete DMEM medium and, after 12 hr, 50 µM Cisplatin (MCE MedChemExpress) was added to the media and left overnight. BMDMs were then incubated with apoptotic cancer cells at a 1:2 ratio and cultured at 37°C for different time points in DMEM supplemented with 10% FBS.

Flow cytometry data were acquired using a BD LSR FORTESSA FACS instrument with FACS DIVA software (BD Biosciences) and analysed using FlowJo V10.4 software. A minimum of $2 \times 10^5$ events were acquired and analysed. Data analysis and interpretation were done using FlowJo software (FlowJo V10.4).

### Image analysis

Confocal images were post-processed and analysed using Fiji distribution of ImageJ version 1.53q. Cell shape descriptors such as 'aspect ratio' (AR), 'circularity' (C) and 'cell area' were measured using Fiji. Specifically, aspect ratio is calculated as (major axis × minor axis − 1) therefore representing solely the degree of elongation, whereas circularity is calculated as [4π*(area × perimeter − 2)], thus representing the degree of similarity to a circumference with a value ranging from 0 to 1 (perfect circle).

### Histology

Tissue was fixed using 4% formaldehyde for 48 hr, dehydrated and paraffin-embedded. Sections (3 µm) were cut and stained using hematoxylin–eosin. For immunodetection, citrate pH 6 buffer was used for antigen retrieval. Staining was used using the following primary antibodies: CD68 (ab125212, Abcam, 1:200), Cathepsin B (12216-1-AP, Proteintech, 1:100), and Cathepsin D (21327-1-AP, Proteintech, 1:400). Dako EnVision+ System HRP labelled Polymer secondary antibodies (Dako) were used, and DAB+ Substrate Chromogen System (Dako) was used for colour development.

For the quantification of loose chromatin remnants in paw oedema images, the pathologist assigned a score ranging from 1 to 3 based on the chromatin content within the inflammatory abscess. A score of 1 indicates low chromatin content, occupying less than 30% of the abscess area, while a score of 3 represents high chromatin content, covering over 60% of the abscess area.

### Statistical analysis

All experiments were conducted with at least three independent biological replicates, each containing technical triplicates. For statistical analysis, we used the Mann–Whitney test to compare control and RBD groups as a non-parametric approach suitable for two independent groups. Additionally, two-way ANOVA was used to evaluate interaction effects between factors in our experimental design, enabling comparisons across multiple groups and conditions.

### Materials availability statement

All materials used in this study are available upon request. Researchers may contact Dr. E. Castellano (ecastellano@usal.es) for access. Most materials have no restrictions; however, access to the mouse model may require a Material Transfer Agreement (MTA) to protect intellectual property or for regulatory compliance. Dr. Castellano can provide details on terms and facilitate the transfer process.

## Acknowledgements

This work was supported by grants from the Spanish Ministry of Science and Innovation (RTI2018-099161-A-I00), Programa JAE-Intro ICU from CSIC (JAEICU-21-IBMCC-6), JCyL (CSI185-20), Marie Curie Initial Training Network on Tumour Infiltrating Myeloid Cell Compartment (PF7 MCA-ITN317445), and CRUK-Barts Cancer Centre Development Fund. This research was co-financed by FEDER funds. The CIC is supported by the Programa de Apoyo a Planes Estratégicos de Investigación de Estructuras de Investigación de Excelencia of Castilla y León autonomous government (CLC-2017-01) and AECC Excellence Program Stop Ras Cancers (EPAEC222641CICS). The authors wish to thank the Pathology Unit, the Mouse Model Experimentation Unit, and the Advanced Cellular Analysis Unit at CIC for their assistance in carrying out this work.

## Additional information

### Funding

| Funder | Grant reference number | Author |
|---|---|---|
| Consejo Superior de Investigaciones Científicas | JAEICU-21-IBMCC-6 | Alejandro Rosell |
| Junta de Castilla y León | CSI185-20 | Marta Alcón Pérez |
| Marie Curie | PF7 MCA-ITN317445 | Agata Adelajda Krygowska |
| Ministerio de Ciencia e Innovación | RTI2018-099161-A-I00 | Alejandro Rosell Cristina Cuesta Juan de Paz Esther Castellano Sanchez |
| Asociación Española Contra el Cáncer | EPAEC222641CICS | Juan de Paz |
| Junta de Castilla y León | CLC-2017-01 | Esther Castellano Sanchez |
| CRUK Barts Centre | Development Fund | Esther Castellano Sanchez |

The funders had no role in study design, data collection and interpretation, or the decision to submit the work for publication.

### Author contributions

Alejandro Rosell, Formal analysis, Investigation, Methodology; Agata Adelajda Krygowska, Conceptualization, Investigation, Methodology; Marta Alcón Pérez, Cristina Cuesta, Mathieu-Benoit Voisin, Juan de Paz, Ottilie Swinyard, Investigation, Methodology; Héctor Sanz-Fraile, Vinothini Rajeeve, Ana Carreras-González, Enrique Gabandé-Rodríguez, Jordi Alcaraz, Juan Anguita, Methodology; Alberto Berral-González, Carmen García-Macías, Javier De Las Rivas, Pedro R Cutillas, Formal analysis; Julian Downward, Resources; Esther Castellano Sanchez, Conceptualization, Funding acquisition, Investigation, Methodology, Writing - original draft, Writing - review and editing

### Author ORCIDs

Mathieu-Benoit Voisin ⓘ https://orcid.org/0000-0003-3001-0894
Vinothini Rajeeve ⓘ https://orcid.org/0000-0002-6361-4291
Enrique Gabandé-Rodríguez ⓘ https://orcid.org/0000-0002-9715-8714
Pedro R Cutillas ⓘ https://orcid.org/0000-0002-3426-2274
Esther Castellano Sanchez ⓘ https://orcid.org/0000-0002-8449-4081

### Ethics

All animal experiments were conducted in accordance with European (2007/526/CE) and Spanish (RD 1201/2005 and RD 53/2013) regulations for the care and use of laboratory animals. Mice were housed, handled, and sacrificed at the NUCLEUS animal facility of the University of Salamanca under standardized conditions. All procedures were approved by the Bioethics Committee of the Cancer Research Center and were performed in compliance with the guidelines and humane endpoints outlined in our approved animal experimental license to minimize pain, suffering, and distress.

Reviewer #2 (Public review): https://doi.org/10.7554/eLife.94590.4.sa1
Author response https://doi.org/10.7554/eLife.94590.4.sa2

## Additional files

### Supplementary files

Supplementary file 1. List of proteins differentially expressed in the secretome of unstimulated *Pik3ca^{RBD/-}* versus *Pik3ca^{WT/-}* bone marrow-derived macrophages (BMDMs).

Supplementary file 2. List of proteins differentially expressed in the secretome of *Pik3ca^{RBD/-}* versus *Pik3ca^{WT/-}* bone marrow-derived macrophages (BMDMs) during apoptotic cell phagocytosis.

MDAR checklist

## Data availability

The mass spectrometry proteomics data have been deposited to the ProteomeXchange Consortium via the PRIDE partner repository with the dataset identifier PXD057794. All data generated or analysed during this study are included in the manuscript and supporting files; source data files have been provided.

The following dataset was generated:

| Author(s) | Year | Dataset title | Dataset URL | Database and Identifier |
|-----------|------|---------------|-------------|------------------------|
| Sanchez C | 2025 | RAS-p110α signalling pathway in macrophages and its impact on both the initiation and resolution of inflammation | https://doi.org/10.6019/PXD057794 | ProteomeXchange, 10.6019/PXD057794 |

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
