## [Editor Report · eLife Assessment]

This **useful** study investigates the impact of disrupting the interaction of RAS with the PI3K subunit p110α in macrophage function in vitro and inflammatory responses in vivo. **Solid** data overall supports a role for RAS-p110α signalling in regulating macrophage activity and so inflammation, however for many of the readouts presented the magnitude of the phenotype is not particularly pronounced. Further analysis would be required to substantiate the claims that RAS-p110α signalling plays a key role in macrophage function. Of note, the molecular mechanisms of how exactly p110α regulates the functions in macrophages have not yet been established.

---

## [Referee Report · Reviewer #2 (Public review)]

Summary:

Cell intrinsic signaling pathways controlling the function of macrophages in inflammatory processes, including in response to infection, injury or in the resolution of inflammation are incompletely understood. In this study, Rosell et al. investigate the contribution of RAS-p110α signaling to macrophage activity. p110α is a ubiquitously expressed catalytic subunit of PI3K with previously described roles in multiple biological processes including in epithelial cell growth and survival, and carcinogenesis. While previous studies have already suggested a role for RAS-p110α signaling in macrophage function, the cell intrinsic impact of disrupting the interaction between RAS and p110α in this central myeloid cell subset is not known.

Strengths:

Exploiting a sound previously described genetically engineered mouse model that allows tamoxifen-inducible disruption of the RAS-p110α pathway and using different readouts of macrophage activity in vitro and in vivo, the authors provide data consistent with their conclusion that alteration in RAS-p110α signaling impairs various but selective aspects of macrophage function in a cell-intrinsic manner.

Weaknesses:

My main concern is that for various readouts, the difference between wild-type and mutant macrophages in vitro or between wild-type and Pik3caRBD mice in vivo is modest, even if statistically significant. To further substantiate the extent of macrophage function alteration upon disruption of RAS-p110α signaling and its impact on the initiation and resolution of inflammatory responses, the manuscript would benefit from a more extensive assessment of macrophage activity and inflammatory responses in vivo.

In the in vivo model, all cells have disrupted RAS-p100α signaling, not only macrophages. Given that other myeloid cells besides macrophages contribute to the orchestration of inflammatory responses, it remains unclear whether the phenotype described in vivo results from impaired RAS-p100α signaling within macrophages or from defects in other haematopoietic cells such as neutrophils, dendritic cells, etc.

Inclusion of information on the absolute number of macrophages, and total immune cells (e.g. for the spleen analysis) would help determine if the reduced frequency of macrophages represents an actual difference in their total number or rather reflects a relative decrease due to an increase in the number of other/s immune cell/s.

Comments on revisions:

I thank the authors for addressing my comments.

- I believe that additional in vivo experiments, or the inclusion of controls for the specificity of the inhibitor, which the authors argue are beyond the scope of the current study, are essential to address the weaknesses and limitations stated in my current evaluation.

- While the neutrophil depletion suggests neutrophils are not required for the phenotype, there are multiple other myeloid cells, in addition to macrophages, that could be contributing or accounting for the in vivo phenotype observed in the mutant strain (not macrophage specific).

- Inclusion of absolute cell numbers (in addition to the %) is essential. I do not understand why the authors are not including these data. Have they not counted the cells?

- Lastly, inclusion of representatives staining and gating strategies for all immune profiling measurements carried out by flow cytometry is important. This point has not been addressed, not even in writing.

---

## [Author Response]

The following is the authors’ response to the current reviews.

Comments on revisions:I thank the authors for addressing my comments.- I believe that additional in vivo experiments, or the inclusion of controls for the specificity of the inhibitor, which the authors argue are beyond the scope of the current study, are essential to address the weaknesses and limitations stated in my current evaluation.

We respectfully acknowledge the reviewer's concern but would like to reiterate that demonstrating the specificity of the inhibitor is beyond the scope of this study. Alpelisib (BYL-719) is a clinically approved drug widely recognized as a specific inhibitor of p110α, primarily used in the treatment of breast cancer. Its selectivity for the p110α isoform has been extensively validated in the literature.

In our study, we used Alpelisib to assess whether pharmacological inhibition of p110α would produce effects similar to those observed in our genetic model, which is particularly relevant for the potential translational implications of our findings. Given the well-documented specificity of this inhibitor, we believe that additional controls to confirm its selectivity are unnecessary within the context of this study. Instead, our focus has been to investigate the functional role of p110α activity in macrophage-driven inflammation using the models described.

We appreciate the reviewer’s insight and hope this clarification addresses their concern.

- While the neutrophil depletion suggests neutrophils are not required for the phenotype, there are multiple other myeloid cells, in addition to macrophages, that could be contributing or accounting for the in vivo phenotype observed in the mutant strain (not macrophage specific).

We appreciate the reviewer's observation regarding the potential involvement of other myeloid cells. However, it is important to highlight that the inflammatory process follows a well-characterized sequential pattern. Our data clearly demonstrate that in the paw inflammation model:

· Neutrophils are effectively recruited, as evidenced by the inflammatory abscess filled with polymorphonuclear cells.

· However, macrophages fail to be recruited in the RBD model.

Given that this critical step is disrupted, it is reasonable to expect that any subsequent steps in the inflammatory cascade would also be affected. A precise dissection of the role of other myeloid populations would require additional lineage-specific models to selectively target each subset, which, as we have previously stated, would be the focus of an independent study.

While we cannot entirely exclude the contribution of other myeloid cells, our data strongly support the conclusion that macrophages are, at the very least, a key component of the observed phenotype. We explicitly address this point in the Discussion section, where we acknowledge the potential involvement of other myeloid populations.

- Inclusion of absolute cell numbers (in addition to the %) is essential. I do not understand why the authors are not including these data. Have they not counted the cells?

We appreciate the reviewer’s concern regarding the inclusion of absolute cell numbers. However, as stated in the Materials and Methods section, we analyzed 50,000 cells per sample, and the percentages reported in the manuscript are directly derived from this standardized count.

Our decision to present the data as percentages follows standard practices in flow cytometry-based analyses, as it allows for a clearer and more biologically relevant comparison of relative changes between conditions. This approach ensures consistency across samples and facilitates the interpretation of population dynamics during inflammation.

We would also like to clarify that all data are based on actual counts, and rigorous controls were implemented throughout the study to ensure accuracy and reproducibility. We hope this explanation addresses the reviewer’s concern and provides further clarity on our approach.

- Lastly, inclusion of representatives staining and gating strategies for all immune profiling measurements carried out by flow cytometry is important. This point has not been addressed, not even in writing.

We appreciate the reviewer’s concern regarding the inclusion of absolute cell numbers. However, as stated in the Materials and Methods section, we analyzed 50,000 cells per sample, and the percentages reported in the manuscript are directly derived from this standardized count.

Our decision to present the data as percentages follows standard practices in flow cytometry-based analyses, as it allows for a clearer and more biologically relevant comparison of relative changes between conditions. This approach ensures consistency across samples and facilitates the interpretation of population dynamics during inflammation.

We would also like to clarify that all data are based on actual counts, and rigorous controls were implemented throughout the study to ensure accuracy and reproducibility. We hope this explanation addresses the reviewer’s concern and provides further clarity on our approach.

The following is the authors’ response to the original reviews.

**Public Reviews:**

**Reviewer #1 (Public review):**
This study by Alejandro Rosell et al. reveals the immunoregulatory role of the RAS-p110α pathway in macrophages, specifically in regulating monocyte extravasation and lysosomal digestion during inflammation. Disrupting this pathway, through genetic tools or pharmacological intervention in mice, impairs the inflammatory response, leading to delayed resolution and more severe acute inflammation. The authors suggest that activating p110α with small molecules could be a potential therapeutic strategy for treating chronic inflammation. These findings provide important insights into the mechanisms by which p110α regulates macrophage function and the overall inflammatory response.The updates made by the authors in the revised version have addressed the main points raised in the initial review, further improving the strength of their findings.
**Reviewer #2 (Public review):**
Summary:Cell intrinsic signaling pathways controlling the function of macrophages in inflammatory processes, including in response to infection, injury or in the resolution of inflammation are incompletely understood. In this study, Rosell et al. investigate the contribution of RAS-p110α signaling to macrophage activity. p110α is a ubiquitously expressed catalytic subunit of PI3K with previously described roles in multiple biological processes including in epithelial cell growth and survival, and carcinogenesis. While previous studies have already suggested a role for RAS-p110α signaling in macrophage function, the cell intrinsic impact of disrupting the interaction between RAS and p110α in this central myeloid cell subset is not known.Strengths:Exploiting a sound previously described genetically engineered mouse model that allows tamoxifen-inducible disruption of the RAS-p110α pathway and using different readouts of macrophage activity in vitro and in vivo, the authors provide data consistent with their conclusion that alteration in RAS-p110α signaling impairs various but selective aspects of macrophage function in a cell-intrinsic manner.Weaknesses:My main concern is that for various readouts, the difference between wild-type and mutant macrophages in vitro or between wild-type and Pik3caRBD mice in vivo is modest, even if statistically significant. To further substantiate the extent of macrophage function alteration upon disruption of RAS-p110α signaling and its impact on the initiation and resolution of inflammatory responses, the manuscript would benefit from a more extensive assessment of macrophage activity and inflammatory responses in vivo.

Thank you for raising this point. We understand the reviewer’s concern regarding the modest yet statistically significant differences observed between wild-type and mutant macrophages in vitro, as well as between wild-type and Pik3ca^RBD^ mice in vivo. Our current study aimed to provide a foundational exploration of the role of RAS-p110α signaling in macrophage function and inflammatory response, focusing on a set of core readouts that demonstrate the physiological relevance of this pathway. While a more extensive in vivo assessment could offer additional insights into macrophage activity and the nuanced effects of RAS-p110α disruption, it would require an array of new experiments that are beyond the current scope.

However, we believe that the current data provide significant insights into the pathway’s role, highlighting important alterations in macrophage function and inflammatory processes due to RAS-p110α disruption. These findings lay the groundwork for future studies that can build upon our results with a more comprehensive analysis of macrophage activity in various inflammatory contexts.

In the in vivo model, all cells have disrupted RAS-p100α signaling, not only macrophages. Given that other myeloid cells besides macrophages contribute to the orchestration of inflammatory responses, it remains unclear whether the phenotype described in vivo results from impaired RAS-p100α signaling within macrophages or from defects in other haematopoietic cells such as neutrophils, dendritic cells, etc.

Thank you for raising this point. To address this, we have added a paragraph in the Discussion section acknowledging that RAS-p110α signaling disruption affects all hematopoietic cells (lines 461-470 in the discussion). However, we also provide several lines of evidence that support macrophages as the primary cell type involved in the observed phenotype. Specifically, we note that neutrophil depletion in chimera mice did not alter transendothelial extravasation, and that macrophages were the primary cell type showing significant functional defects in the paw edema model. These findings, combined with specific deficiencies in myeloid populations, suggest a predominant role of macrophages in the impaired inflammatory response, though we acknowledge the potential contributions of other myeloid cells.

Inclusion of information on the absolute number of macrophages, and total immune cells (e.g. for the spleen analysis) would help determine if the reduced frequency of macrophages represents an actual difference in their total number or rather reflects a relative decrease due to an increase in the number of other/s immune cell/s.

Thank you for this suggestion. We understand the value of presenting actual measurements; however, we opted to display normalized data to provide a clearer comparison between WT and RBD mice, as this approach highlights the relative differences in immune populations between the two groups. Normalizing data helps to focus on the specific impact of the RAS-p110α disruption by minimizing inter-sample variability that can obscure these differences.

To further address the reviewer’s concern regarding the interpretation of macrophage frequencies, we have included a pie chart that represents the relative proportions of the various immune cell populations studied within our dataset. Author response image 1 provides a visual overview of the immune cell distribution, enabling a clearer understanding of whether the observed decrease in macrophage frequency represents an actual reduction in total macrophage numbers or a shift in their relative abundance due to changes in other immune populations.

We hope this approach satisfactorily addresses reviewer’s concerns by providing both a normalized dataset for clearer interpretation of genotype-specific effects and an overall immune profile that contextualizes macrophage frequency within the broader immune cell landscape.

**Author response image 1. sa2fig1:** 

**Recommendations for the authors:**

**Reviewer #2 (Recommendations for the authors):**
As proof of concept data that activation of RAS-p110α signaling constitutes indeed a putative approach for treating chronic inflammation is not included in the manuscript, I suggest removing this implication from the abstract.

Thank you for this suggestion. We have now removed this implication from the abstract to maintain clarity and to better reflect the scope of the data presented in the manuscript.

Inclusion of a control in which RBD/- cells are also treated with BYL719, across experiments in which the inhibitor is used, would be important to determine, among other things, the specificity of the inhibitor.

We appreciate the reviewer’s suggestion to include RBD/- cells treated with BYL719 as an additional control. However, we would like to clarify that this approach would raise a different biological question, as treating RBD mice with BYL719 would not only address the specificity of the inhibitor but also examine the combined effects of genetic and pharmacologic disruptions on PI3K pathway signaling. Investigating this dual disruption falls outside the scope of our current study, which is focused specifically on the effects of RAS-p110α disruption.

It is also important to note that our RBD mouse model selectively disrupts RAS-mediated activation of p110α, while PI3K activation can still occur through other pathways, such as receptor tyrosine kinases (RTKs) and G protein-coupled receptors (GPCRs). Thus, inhibiting p110α with BYL719 would produce broader effects beyond the inhibition of RAS-PI3K signaling, impacting PI3K activation regardless of its upstream source.

In addition, incorporating this control would require us to repeat nearly all experiments in the manuscript, as it would necessitate generating and analyzing new samples for each experimental condition. Given the scope and resources involved, we believe this approach is unfeasible at this stage of the revision process.

We hope this explanation is satisfactory and that the current data in the manuscript provide a rigorous assessment of the RAS-p110α signaling pathway within the defined experimental scope.

Figure 3I is missing the statistical analysis (this is mentioned in the legend though).

Thank you for pointing this out. We apologize for the oversight. The statistical analysis for Figure 3I has now been added.

Gating strategies and representative staining should be included more generally across the manuscript.

Thank you for this suggestion. To address this, we have added a new supplementary figure (Figure 2-Supplement Figure 2) that illustrates the gating strategy along with a representative dataset. Additionally, a brief summary of the gating strategy has been included in the main text to further clarify the methodology.

It is recommended that authors show actual measurements rather than only data normalized to the control (or arbitrary units).

Thank you for this suggestion. We understand the value of presenting actual measurements; however, we opted to display normalized data to provide a clearer comparison between WT and RBD mice, as this approach highlights the relative differences in immune populations between the two groups. Normalizing data helps to focus on the specific impact of the RAS-p110α disruption by minimizing inter-sample variability that can obscure these differences.

To further address the reviewer’s concern regarding the interpretation of macrophage frequencies, we have included a pie chart that represents the relative proportions of the various immune cell populations studied within our dataset. Author response image 1 provides a visual overview of the immune cell distribution, enabling a clearer understanding of whether the observed decrease in macrophage frequency represents an actual reduction in total macrophage numbers or a shift in their relative abundance due to changes in other immune populations.

We hope this approach satisfactorily addresses reviewer’s concerns by providing both a normalized dataset for clearer interpretation of genotype-specific effects and an overall immune profile that contextualizes macrophage frequency within the broader immune cell landscape.